# DOMAIN2VEC: VECTORIZING DATASETS TO FIND THE OPTIMAL DATA MIXTURE WITHOUT TRAINING

## ABSTRACT

The mixture ratio of data from different source domains significantly affects the performance of language models (LM) pretraining. In this paper, we introduce DOMAIN2VEC, a novel approach that decomposes any dataset into a linear combination of several "Meta-Domains", a new concept designed to capture key underlying features of datasets. DOMAIN2VEC maintains a vocabulary of Meta-Domains and uses a Meta-Domain Classifier to decompose any given dataset into a domain vector that corresponds to a distribution over this vocabulary. These domain vectors enable the identification of optimal data mixture ratio for LM pretraining in a training-free manner under the *Distribution Alignment Assumption* ($DA^2$), which suggests that when the data distribution of the training set and the validation set is more aligned, a lower validation loss is achieved. Moreover, previous work could use DOMAIN2VEC to model the relationship between domain vectors and LM performance, greatly enhancing the scalability of previous methods without retraining as new datasets are introduced. Extensive experiments demonstrate that DOMAIN2VEC finds data mixture ratios that enhance downstream task performance with minimal computational overhead. Specifically, DOMAIN2VEC achieves the same validation loss on Pile-CC using only $51.5\%$ of the compute required when training on the original mixture of The Pile Dataset. Under equivalent compute budget, DOMAIN2VEC improves downstream performance by an average of $2.72\%$. DOMAIN2VEC serves as a strong and efficient baseline for data mixture optimization in LM pretraining, offering insights into improving data efficiency in large-scale models.

## 1 INTRODUCTION

Through training on large-scale text corpora, Large Language Models (LLMs) have demonstrated strong generalization capabilities (Touvron et al., 2023; OpenAI et al., 2024; Yang et al., 2024; DeepSeek-AI et al., 2024). Training datasets for LLMs are typically divided into multiple domains based on their sources. For example, a widely used dataset, The Pile (Gao et al., 2021), includes $12.07\%$ Books3, $8.96\%$ ArXiv, $6.12\%$ FreeLaw, etc. Recent studies have highlighted that mixture proportions of different domains (referred to as data mixture) could significantly impact the effectiveness of language models (Hoffmann et al., 2022a; Xie et al., 2023b), with data from one domain potentially influencing the outcomes of others (Guo et al., 2022). Typically, the data mixture used for training large language models are determined heuristically or based on downstream performance metrics, which is often unscalable and may lead to suboptimal mixtures. Thus, finding the optimal data mixture in a scalable and efficient manner is a critical research question (Liu et al., 2024).

Recently, researchers have proposed various methods to predict the optimal data mixture. In this paper, we categorize prior work into two lines. The first line **implicitly adjusts** the data mixture via finding high-quality data from different domains or datasets. Lin et al. (2024) propose using Selective Language Models to select useful tokens to align with the ideal data mixture. Ankner et al. (2024) and Thakkar et al. (2023) directly filter out some low-quality data at the sample level based on the perplexity or the influence score. The second line of work focuses more on modeling the relationship between the data mixture and the performance of language models, which **explicitly adjusts** the data mixture of different domains or datasets. A straightforward method is to train language models on different data mixtures and select the one that yields the best performance, as seen in the training of Gopher (Rae et al., 2022). However, it is impossible to enumerate all possible data

mixtures owing to the enormous computation costs. To address this issue, Xie et al. (2023a) propose DoReMi, which leverages a well-trained reference model to guide the training of another proxy model using Group DRO (Nemirovski et al., 2009; Sagawa* et al., 2020) over different datasets. The optimized data mixture derived from this process is then used to train a large model. While DoReMi enhances training efficiency for identifying better data mixtures, it still relies heavily on having a well-trained reference model, and it remains difficult to determine what qualifies as a good reference model. To reduce this dependence, Fan et al. (2023) introduce DoGE, which assigns greater weight to a domain based on its contribution to the learning of target domains. Inspired by scaling laws Kaplan et al. (2020); Hoffmann et al. (2022b), to build a functional relationship between data mixture and the performance of language models rather than providing a single data mixture (Xie et al., 2023a; Fan et al., 2023), several works (Ye et al., 2024; Ge et al., 2024; Gu et al., 2024; Que et al., 2024) attempt to fit nonlinear expressions through extensive experiments on smaller proxy models. Gu et al. (2024) also accurately predicted that the pretrained domain loss would first rise and then fall during continue pretraining, and introduced the critical mixture ratio to mitigate catastrophic forgetting on the pretrained domain. Instead of using nonlinear expressions, Liu et al. (2024) propose RegMix, which formulates the search for optimal data mixture as a regression task and fits a regression model to predict the performance of different data mixture.

While prior work has shown promising results, they have some issues as follows: **1) Higher Computational Cost**: For instance, although the proxy model used in DoReMi (Xie et al., 2023a) has only 280M parameters, the estimated FLOPs of DoReMi is $3.7 \times 10^{19}$. Similarly, RHO-1 (Lin et al., 2024) only calculates loss on certain tokens but still requires the entire sentence to be input into the model. **2) Lack of Scalability** : When building the functional relationship like Ye et al. (2024) and Liu et al. (2024), the dimension of the independent variable (i.e., the number of different datasets) is fixed. If we change components of training dataset (i.e., introduce some new datasets, filter some low-quality data), the previously fitted functions cannot be generalized to current datasets. This necessitates resampling new data mixtures, retraining proxy models, and refitting the functions, which severely limits the scalability of these methods.

To address these issues, we introduce DOMAIN2VEC, a newly introduced concept to capture the underlying features of datasets. DOMAIN2VEC maintains a vocabulary of "Meta-Domains". We hypothesize that *any dataset, regardless of its source, be approximated by a linear combination of several Meta-Domains in certain distribution*. This distribution over the vocabulary could serve as the vector representation (or domain vector) of the current dataset. To efficiently determine which Meta-Domains comprise a given dataset, we propose utilizing a Meta-Domain Classifier to generate the domain vector and outline a concrete pipeline to build a Meta-Domain Classifier from scratch. For finding the optimal data mixture for language model pretraining, we introduce the *Distribution Alignment Assumption* (DA$^2$), stating that *lower validation loss can be achieved when the domain vector of training datasets aligns with domain vector of the validation datasets*. Instead of modeling the relationship between data mixture and language model performance like previous work (Liu et al., 2024; Ye et al., 2024; Que et al., 2024), we focus on modeling the relationship between domain vectors provided by DOMAIN2VEC and the LM performance which significantly enhances the scalability of prior methods. Notably, regardless of changes to the training datasets, DOMAIN2VEC could still provide corresponding domain vector. Moreover, combining different datasets is equivalent to combining their respective domain vectors. This allows us to predict the performance of various data mixtures without the need to retrain proxy models to fit these fictional relationship again, further improving efficiency.

In summary, we highlight our contributions as follows:

1. We propose DOMAIN2VEC, a novel concept to capture the underlying features of datasets. We also propose viewing datasets as combinations of "Meta-Domains" and propose an efficient pipeline for vectorizing a dataset using a Meta-Domain Classifier.

2. We propose *Distribution Alignment Assumption* (DA$^2$) for language model pretraining, a training-free method to identify the optimal data mixture. Additionally, we demonstrate how to integrate DOMAIN2VEC into prior work, which greatly enhances the scalability of prior work without retraining as training datasets changes.

3. We validate the effectiveness of DOMAIN2VEC from two aspects: text generation ability and downstream task performance. Experimental results show that our method could accurately predict the performance of different data mixtures without the need for training any

proxy model. Moreover, we identify data mixtures that achieve downstream performance close to DoReMi (Xie et al., 2023a), while using only $0.26\%$ of its computational cost.

## 2 DOMAIN2VEC

In this section, we introduce DOMAIN2VEC, an algorithm that decomposes a dataset into a linear combination of various "Meta-Domains". This approach allows us to represent the underlying features of datasets through a normalized vector. We also outline a pipeline for constructing the vocabulary of DOMAIN2VEC and training a Meta-Domain classifier.

**Key Assumption**  DOMAIN2VEC maintains a vocabulary, a set of "Meta-Domains". Assume we have $n$ Meta-Domains $\mathcal{D}_j^*$ $(0 \leq j < n)$, where $\mathcal{D}_j^*$ is represented as $\boldsymbol{e}_j$, a one-hot vector where the $j$-th element is $1$. We hypothesize that, for any given dataset $\mathcal{D}$, it could be represented as a domain vector $\boldsymbol{v}$, by linear combination of these Meta-Domains. Specifically,

$$\boldsymbol{v} \approx \sum_{j=0}^{n-1} v_j \cdot \boldsymbol{e}_j, \tag{1}$$

where each element $v_j$ of $\boldsymbol{v}$ represents the projection (weight) of the dataset $\mathcal{D}$ on $\mathcal{D}_j^*$. Thus, $\boldsymbol{v} = [v_0, v_1, v_2, ..., v_{n-1}]^\top$ can be a representation (distribution) of the dataset $\mathcal{D}$ over the Meta-Domains.

**Construct the Vocabulary of DOMAIN2VEC**  First, we argue that constructed Meta-Domains, which could represent dataset from any source, requires satisfying these following three conditions:

1. The original data for constructing Meta-Domains should be as diverse and large as possible.
2. The method for constructing Meta-Domains should be computationally efficient.
3. There should be distinct differences between different Meta-Domains.

We collected data from more than $100$ sources across three coarse domains: English, Chinese, and Code. After deduplication, we obtained around $5.2$TB of text data. First, we utilize bge-small-en-v1.5 and bge-small-zh-v1.5 (Xiao et al., 2023) to compute embeddings for the English and Chinese data, respectively. Then, we employ K-Means (Macqueen, 1967; Arthur & Vassilvitskii, 2006) to cluster these embeddings, resulting in $240$ different Meta-Domains for English and Chinese Data. We also demonstrate the relationship between the number of Meta-Domain and inertia (measuring the distance between each data point and its centroid) in Figure 1. As for the Code data, we directly classified these data based on their programming language categories, ultimately constructing $20$ Meta-Domains for code, covering mainstream programming languages. **Finally, we construct $260$ unique Meta-Domains**.

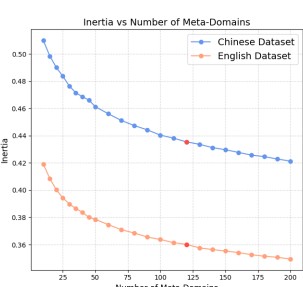

Figure 1: The relationship between the number of Meta-Domains and Inertia.

**Meta-Domain Classifier**  In this section, we will introduce how to obtain the normalized domain vector for any given dataset $\mathcal{D}_i$, which satisfies Equation 1. First, we trained a Meta-Domain Classifier based on Qwen2-1.5b-base (Yang et al., 2024). For any given text $text_j \in \mathcal{D}_i$, we have

$$\boldsymbol{p}_j = [p_0, p_1, p_2, ..., p_{n-1}]^\top = \text{Classifier}(text_j) \tag{2}$$

where $p_i$ represents the probability that $text_j$ belongs to the $i$-th Meta-Domain. For $\mathcal{D}_i$, we could sample $N$ texts then take the average of domain vector of these samples. Thus, the domain vector $\boldsymbol{v}_i$ of dataset $\mathcal{D}_i$ is,

$$\boldsymbol{v}_i \approx \frac{1}{N} \sum_{j=0}^{N-1} \boldsymbol{p}_j \tag{3}$$

Then, we could use the vector $\boldsymbol{v}_i$ to approximately represent the feature of dataset $\mathcal{D}_i$ from any source. Meanwhile, during the pretraining phase of large language models, we typically have training datasets $\mathcal{D}_{train} = \{\mathcal{D}_1, \mathcal{D}_2, ..., \mathcal{D}_k\}$ from multiple sources. We can convert each of these datasets into domain vectors following Equation 2 and 3. Therefore, $\mathcal{D}_{train}$ can be approximately represented as $\boldsymbol{V}_{train} = [\boldsymbol{v}_1, \boldsymbol{v}_2, ..., \boldsymbol{v}_k]$, where $\boldsymbol{V}_{train} \in \mathbb{R}^{k \times n}$.

**Training and Evaluation** The Meta-Domain Classifier is trained to determine which Meta-Domain an arbitrary text from the training set originally belongs to. Thus, we extract $3,000$ texts from each meta-domain for training and $500$ documents for evaluation. We add a classifier head to Qwen2-1.5B-base (Yang et al., 2024), which has a shape of (hidden size, 260), where 260 equals the number of Meta-Domains. Then, we use the Adam (Kingma & Ba, 2017) optimizer with a learning rate of 2e-5 and train the classifier for 3 epochs via cross entropy loss. After that, we evaluate the performance of the meta-domain classifier on the test set, achieving a classification accuracy of $74.73\%$. Meanwhile, we also sample $1,000$ examples from each sub-dataset of The Pile (Gao et al., 2021). Following Equation 3, we obtain domain vectors predicted by the Meta-Domain classifier for each sub-dataset, as shown in Figure 2. It can be seen that the distribution of the Pile's sub-datasets over the meta-domains is very different. This phenomena not only indicates that **our classifier could reasonably distinguish some base features from different datasets**, but also demonstrates that the various meta-domains have significant semantic differences.

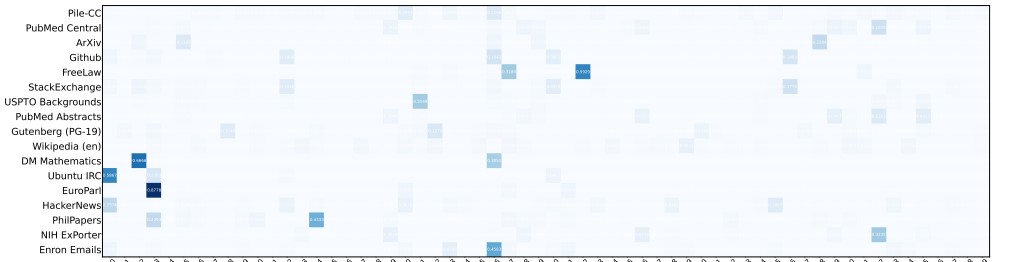

Figure 2: The Domain Vector of each sub-dataset of The Pile (Gao et al., 2021), where each row corresponds to a sub-dataset and each column corresponds to a Meta-Domain. The higher the proportion of data belonging to a particular Meta-Domain, the closer the color of the corresponding cell is to blue). We only display the distribution on some English Meta-Domains for clarity. The full picture is shown in Figure 7.

## 3 FINDING THE OPTIMAL DATA MIXTURE USING DOMAIN2VEC

In this section, we will introduce how to find the optimal data mixture using DOMAIN2VEC in a training free manner called "*Distribution Alignment Assumption* (DA$^2$)". We will also demonstrate how to incorporate our DOMAIN2VEC tools to prior work, which greatly enhance the scalability of previous work as new datasets are introduced[1].

### 3.1 TASK FORMULATION

During the pretraining phase of large language models, we typically collect training datasets $\mathcal{D}_{train} = \{\mathcal{D}_1, \mathcal{D}_2, ..., \mathcal{D}_k\}$ from multiple sources (e.g., ArXiv, Wikipedia). We also pre-define a validation set $\mathcal{D}_{valid}$, which might be independently and identically distributed with the training dataset or might be unrelated to the training dataset (e.g., data that exceeds the training dataset cutoff date, data with quality but small quantity). Accordingly, the data mixture $\boldsymbol{r} = [r_1, r_2, ..., r_k]^\top, 0 \leq r_i \leq 1, \sum_{i=1}^{k} r_i = 1$ specifies the sampling probability distribution over different trainsets. Let the trained language model be denoted as $\theta$, and the validation loss of the model be denoted as $\mathcal{L}_\theta$. Thus, the optimization objective of finding the optimal data mixture $\boldsymbol{r}^*$ is to improve the performance of language models, such as minimizing the validation loss, as shown in Equation 4. $\mathcal{L}^{\mathcal{D}_{valid}}(\boldsymbol{r})$ represents the validation loss of the language model pretrained via data mixture $\boldsymbol{r}$.

$$\boldsymbol{r}^* = \arg\min_{\boldsymbol{r}}(\min_{\theta} \mathcal{L}_\theta^{\mathcal{D}_{valid}}(\boldsymbol{r})) \triangleq \arg\min_{\boldsymbol{r}} \mathcal{L}^{\mathcal{D}_{valid}}(\boldsymbol{r}) \tag{4}$$

### 3.2 DISTRIBUTION ALIGNMENT ASSUMPTION (DA$^2$)

Empirically, when the data distribution of the training set $\mathcal{D}_{train}$ and the validation set $\mathcal{D}_{valid}$ is consistent, we could achieve a lower validation loss $\mathcal{L}^{\mathcal{D}_{valid}}$ on the validation set[2]. The most essential

---

[1]The pseudo code of DOMAIN2VEC+DA$^2$ and DOMAIN2VEC + RegMix are shown in Appendix A.2.

[2]We also have provided the detailed description in the Appendix A.1.

question is "*How do we model the data distribution of various datasets?*". Fortunately, according to Section 2, for the training dataset $\mathcal{D}_{train}$, we can obtain its vector representation $\boldsymbol{V}_{train} \in \mathbb{R}^{k \times n}$, which models some base features of $\mathcal{D}_{train}$. Correspondingly, for the validation set $\mathcal{D}_{valid}$, we also have its vector representation $\boldsymbol{q}_{valid} = [q_0, q_1, q_2, ..., q_{n-1}]^\top$. After mixing $\mathcal{D}_{train}$ with data mixture $\boldsymbol{r}$, the final distribution over Meta-Domains of $\mathcal{D}_{train}$ is given by $\boldsymbol{V}_{train} \cdot \boldsymbol{r}$. Therefore, based on the distribution alignment assumption, Equation 4 can be equivalently written as:

$$\boldsymbol{r}^* = \arg \min_{\boldsymbol{r}} \mathrm{Dist}(\boldsymbol{V}_{train} \cdot \boldsymbol{r}, \boldsymbol{q}_{valid}) \tag{5}$$

where $\mathrm{Dist}(\cdot, \cdot)$ is a distance function used to measure the similarity between two vectors. In this paper, we use Huber Loss (Huber, 1964; Hastie et al., 2009) to measure the similarity.

### 3.3 Applying Domain2Vec to Prior Work

As mentioned before, we could directly combine DOMAIN2VEC with prior work, which could address the issue of needing to refit the relationship $\mathcal{L}^{\mathcal{D}_{valid}}(\boldsymbol{r})$ between the data mixture $\boldsymbol{r}$ and the validation loss $\mathcal{L}$ as new datasets are introduced. In this section, we will introduce how to integrate DOMAIN2VEC with RegMix (Liu et al., 2024) to search for the optimal data mixture. Following Liu et al. (2024), we train a Linear Regression Model ($\hat{y} = \boldsymbol{\omega}^\top \cdot \boldsymbol{x}$) like Equation 6 to fit $\mathcal{L}(\boldsymbol{p})$ on $\mathcal{D}_i^*$ (notated as $\mathcal{L}^{D_i^*}(\boldsymbol{p})$) to build the relationship between the validation loss on $\mathcal{D}_i^*$ and domain vector $\boldsymbol{p} = [p_0, p_1, p_2, ..., p_{n-1}]^\top, 0 \leq p_i \leq 1, \sum_{i=0}^{n-1} p_i = 1$. Formally,

$$\boldsymbol{\omega}_i^* = \arg \min_{\boldsymbol{\omega}} \|\mathcal{L}^{D_i^*}(\boldsymbol{p}) - \boldsymbol{\omega}^\top \cdot \boldsymbol{p}\| \tag{6}$$

Because any validation set $\mathcal{D}_{valid}$ can also be viewed as a linear combination of multiple Meta-Domains, i.e., $\mathcal{D}_{valid} \approx \sum_{i=0}^{n-1} q_i \cdot \mathcal{D}_i^*$. Meanwhile, the validation loss over different Meta-Domains are additive. Thus, the validation loss on the $\mathcal{D}_{valid}$ of the data mixture $\boldsymbol{p}$ is,

$$\mathcal{L}^{\mathcal{D}_{valid}}(\boldsymbol{p}) = \sum_{i=0}^{n-1} q_i \cdot \mathcal{L}^{D_i^*}(\boldsymbol{p}) = \sum_{i=0}^{n-1} q_i \cdot (\boldsymbol{\omega}_i^*)^\top \cdot \boldsymbol{p} \tag{7}$$

After mixing $\mathcal{D}_{train}$ according to the data mixture $\boldsymbol{r}$, the final distribution over the Meta-Domain is given by $\boldsymbol{V}_{train} \cdot \boldsymbol{r}$. Therefore, we can conclude that $\boldsymbol{p} = \boldsymbol{V}_{train} \cdot \boldsymbol{r}$. Substituting $\boldsymbol{p} = \boldsymbol{V}_{train} \cdot \boldsymbol{r}$, Equation 4 can be equivalently written as follows,

$$\boldsymbol{r}^* = \arg \min_{\boldsymbol{r}} \sum_{i=0}^{n-1} q_i \cdot \mathcal{L}^{\mathcal{D}_i^*}(\boldsymbol{V}_{train} \cdot \boldsymbol{r}) \tag{8}$$

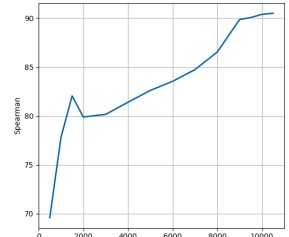

Figure 3: The relationship between the number of trained data mixture and the Spearman Correlation Coefficient.

To fit Equation 6 for each Meta-Domain, we sampled $10, 500$ diverse data mixture from a Dirichlet distribution based on the token distribution of Meta-Domains. Then we used these data mixtures to train different models with 85M parameters on 1B tokens. We used LightGBM to fit Equation 6 for each Meta-Domain. We also reserved data mixtures that were not trained by LightGBM to evaluate whether fitted equations can accurately predict the validation loss for unseen data mixture. The Spearman Correlation Coefficient between the actual loss and the predicted loss by LightGBM is shown in Figure 3. As the trained data mixture increases, the predictions made by LightGBM become more accurate.

## 4 Domain2Vec Helps Find the Optimal Data Mixture with Less Computation, Even Without Training

The motivation for finding the optimal data mixture is to "Enhance the performance of large language models". The performance of large language models can be evaluated from two perspectives: 1) Text generation ability, which refers to the language modeling loss or perplexity on the hold-out validation dataset. 2) Downstream task performance, such as MMLU (Hendrycks et al., 2021) and GSM8K (Cobbe et al., 2021). Therefore, for the text generation ability, we should find the optimal

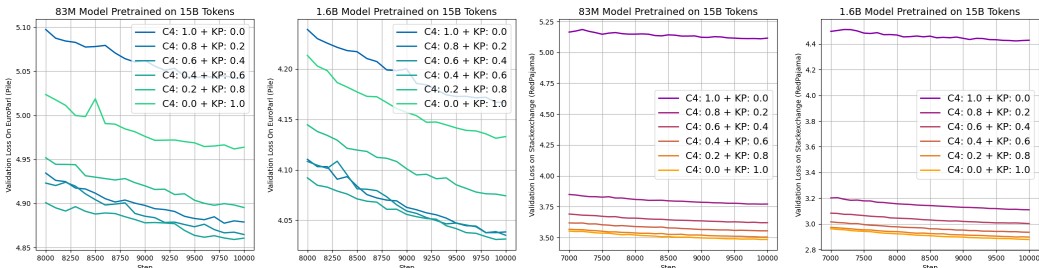

Figure 4: The validation loss on the EuroParl (The Pile) and Stackexchange (RedPajama) of models trained using data mixture in Table 1. The loss on other validation sets are shown in Appendix A.4.

data mixture to minimize the validation loss. For downstream task performance, we should find the optimal data mixture to maximize the downstream task performance. By deploying DOMAIN2VEC, we could accurately predict the validation loss of any training dataset with different mixture ratios on any validation dataset, even without the need for training some proxy models. Moreover, we used only $0.26\%$ of the computational costs required by DoReMi (Xie et al., 2023a) to find a data mixture with performance comparable to baselines like DoReMi.

## 4.1 MINIMIZE THE VALIDATION LOSS

### 4.1.1 PILOT STUDY

Using the validation loss of the large language model as a metric to evaluate its generation capabilities is very straightforward. *However, is there a data mixture that can simultaneously achieve the lowest loss across all validation sets? Can the optimal data mixture generalize across models of different model size?* These questions are essential for the study of data mixture of large language

Table 1: The data mixture we used to mix C4 (Raffel et al., 2020) and Knowledge Pile (Fei et al., 2024).

| Dataset | **Data Mixture** | | | | | |
|---|---|---|---|---|---|---|
| C4 | 0 | 0.2 | 0.4 | 0.6 | 0.8 | 1.0 |
| Knowledge Pile | 1.0 | 0.8 | 0.6 | 0.4 | 0.2 | 0.0 |

models. To answer these questions, we first mix C4 (Raffel et al., 2020) and Knowledge Pile (Fei et al., 2024) with different data mixtures as the training set as shown in Tabel 1. We pretrain two Transformer (Vaswani et al., 2017) Decoder-only models with 83M and 1.6B parameters from scratch using a next-token prediction loss. During pretraining, we evaluate the validation loss of models trained with different mixture ratios on 20 subsets of The Pile (Gao et al., 2021) and RedPajama (Computer, 2023), as shown in Figure 4. We find that, *for different validation sets, the ranking of mixture ratios varies significantly*. For each validation dataset, we also rank all the data mixtures based on their validation loss and calculate the Spearman and Pearson correlation coefficients of the data mixture ranking between the 83M model and the 1.6B model on various validation sets. The Spearman correlation coefficient is $0.9743$, and the Pearson correlation coefficient is $0.9947$. Thus, *for the same validation set, the data mixture ranking of validation loss on identical validation dataset does not change with the variation in model parameters*. This phenomena indicates that we could find the optimal data mixture without the need to train a large model. Based on these findings, we will demonstrate how DOMAIN2VEC could predict the ranking of different mixture ratios even without training some small proxy models.

### 4.1.2 EXPERIMENTAL SETUP

**Dataset & Data Mixture** Colossal Clean Crawled Corpus (C4) (Raffel et al., 2020) is a colossal and cleaned version of Common Crawl's web crawl corpus. Knowledge Pile (Fei et al., 2024) is a high-quality 735 GB dataset which could significantly improves the performance of large language models in knowledge-related and mathematical reasoning tasks. We mix C4 and Knowledge Pile with different data mixtures as the training set as shown in Tabel 1.

**Training Setup** We pretrained some Transformer (Vaswani et al., 2017) Decoder-only models with 83M and 1.6B parameters from scratch using a next-token prediction loss. All the models have a batch size of 1.5M tokens, and the maximum sequence length is 4096. We use the Adam (Kingma

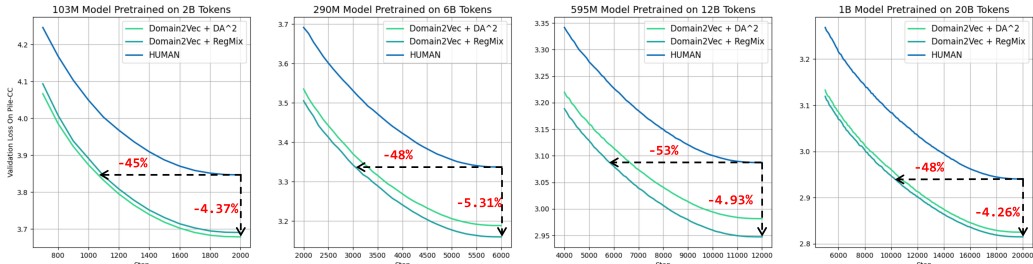

Figure 5: The validation loss on the Pile-CC subset. DOMAIN2VEC achieves the comparable validation loss of Human (The model using original data mixture from The Pile), which only uses almost $51.5\%$ training computational costs of Human. Using the same training cost, DOMAIN2VEC can reduce the validation loss by approximately $4.72\%$ compared to Human.

& Ba, 2017) optimizer with gradient clip of $1.0$. The learning rate linearly warms up to a maximum learning rate of 2e-4 over the first $100$ steps, then decreases to 2e-5 using a cosine learning rate scheduler with 10,000 steps. The detailed parameters of models we used are shown in the Table 6.

**Evaluation** Because the optimal mixture ratio varies for different validation datasets, it is impossible to find a data mixture that is optimal for all validation sets. Therefore, we turn to predict the ranking of loss on 20 validation datasets from The Pile (Gao et al., 2021) and RedPajama (Computer, 2023) for the six different mixture ratios shown in Table 1. Then, we evaluate our proposed method using the Spearman correlation coefficient and the Pearson correlation coefficient between the predicted ranking and the actual ranking.

### 4.1.3 EXPERIMENTAL RESULTS

First, we present the validation loss curves for various data mixtures in Figure 4 and the Appendix A.1. It can be observed that, on most validation sets, incorporating a certain amount of Knowledge Pile significantly reduces the model's validation loss, even on the C4 validation set from RedPajama.

Table 2: The results of deploying the DOMAIN2VEC to predict the ranking of different Validation sets.

| Metrics | Random | DOMAIN2VEC+DA$^2$ | DOMAIN2VEC+RegMix |
|---|---|---|---|
| Pearson | 0.0300 | **0.5833** | 0.3881 |
| Spearman | 0.0497 | **0.6657** | 0.4629 |

This indicates the high quality of the training data in the Knowledge Pile. Then, we sample $10,000$ samples from C4 and Knowledge Pile respectively, and $1,000$ samples from each validation set. After that, we apply DOMAIN2VEC to rank the data mixture, as shown in Table 1. As demonstrated in Table 2, the ranking predicted by DOMAIN2VEC exhibits a strong positive correlation with the actual ranking, significantly outperforming random guessing. Interestingly, we find that DOMAIN2VEC + RegMix even predicted that a mixture of $20\%$ Knowledge Pile and $80\%$ C4 could achieve the lowest validation loss on C4 validation set from RedPajama. We hypothesize that this is due to the higher data quality of Knowledge Pile compared to C4, as well as the overlap between these two datasets in certain Meta-Domains. As a result, incorporating a portion of Knowledge Pile into the mixture likely enhances the training of C4. It is also important to note that our method is a ***training-free approach***, unlike prior works that rely on training small proxy models to rank data mixtures. Despite this more challenging setup, our method accurately predicts the rankings of different data mixtures. We believe these experimental results could offer valuable insights for the community.

### 4.2 MAXIMIZE THE DOWNSTREAM TASK PERFORMANCE.

In this section, we demonstrate how to use DOMAIN2VEC to identify the optimal data mixture for maximizing downstream task performance. A key question is how to model the relationship between data mixture and downstream task performance. Fortunately, Liu et al. (2024) finds that *the validation loss on Pile-CC has the highest correlation with the downstream performance across their evaluations*. To make a comparison with previous work, we use the same evaluation datasets as Liu et al. (2024). Thus, our task is to find a data mixture that minimizes the validation loss on Pile-CC. ***Experimental results reveal that*** DOMAIN2VEC ***predicts a data mixture with performance comparable to DoReMi (Xie et al., 2023a), while using only*** $0.26\%$ ***computational cost***.

Table 3: Downstream Task Performance of different models pretrained on different data mixture. Similiar to Liu et al. (2024), Human refers the original data mixture from The Pile. Pile-CC Only refers only training on the Pile-CC subset. The data mixture and estimated flops of DoReMi and RegMix are from Liu et al. (2024). All the data mixture we used are shown in Table 4 and Table 5. The results of 106M models pretrained on 2B tokens are showin in Table 7 owing to the page limit.

| Benchmark | Human | DoReMi | Pile-CC Only | RegMix | DOMAIN2VEC + DA$^2$ | DOMAIN2VEC + RegMix |
|---|---|---|---|---|---|---|
| *290M Model Pretrained on 6B Tokens* | | | | | | |
| Social IQA | 0.364 | 0.373 | 0.374 | 0.371 | 0.371 | 0.368 |
| HellaSwag | 0.295 | 0.312 | 0.317 | 0.315 | 0.307 | 0.312 |
| PiQA | 0.605 | 0.631 | 0.639 | 0.642 | 0.624 | 0.633 |
| OpenBookQA | 0.261 | 0.271 | 0.271 | 0.262 | 0.268 | 0.266 |
| Lambada | 0.175 | 0.208 | 0.206 | 0.210 | 0.182 | 0.208 |
| SciQ | 0.711 | 0.682 | 0.663 | 0.674 | 0.670 | 0.697 |
| ARC Easy | 0.395 | 0.410 | 0.419 | 0.417 | 0.420 | 0.412 |
| COPA | 0.632 | 0.660 | 0.682 | 0.657 | 0.627 | 0.642 |
| RACE | 0.265 | 0.280 | 0.280 | 0.276 | 0.283 | 0.281 |
| LogiQA | 0.283 | 0.293 | 0.296 | 0.276 | 0.277 | 0.292 |
| WinoGrande | 0.511 | 0.506 | 0.509 | 0.524 | 0.498 | 0.504 |
| MultiRC | 0.507 | 0.555 | 0.513 | 0.545 | 0.521 | 0.517 |
| Average Performance | 0.417 | 0.432 | 0.431 | 0.431 | 0.421 | 0.428 |
| *595M Model Pretrained on 6B Tokens* | | | | | | |
| Social IQA | 0.378 | 0.387 | 0.390 | 0.394 | 0.383 | 0.388 |
| HellaSwag | 0.338 | 0.377 | 0.386 | 0.385 | 0.355 | 0.366 |
| PiQA | 0.624 | 0.656 | 0.663 | 0.667 | 0.651 | 0.659 |
| OpenBookQA | 0.273 | 0.279 | 0.283 | 0.294 | 0.288 | 0.271 |
| Lambada | 0.255 | 0.294 | 0.332 | 0.310 | 0.269 | 0.292 |
| SciQ | 0.777 | 0.757 | 0.770 | 0.791 | 0.763 | 0.769 |
| ARC Easy | 0.439 | 0.453 | 0.478 | 0.481 | 0.453 | 0.460 |
| COPA | 0.642 | 0.680 | 0.672 | 0.663 | 0.668 | 0.667 |
| RACE | 0.289 | 0.309 | 0.311 | 0.311 | 0.288 | 0.303 |
| LogiQA | 0.263 | 0.268 | 0.252 | 0.267 | 0.263 | 0.267 |
| WinoGrande | 0.509 | 0.515 | 0.506 | 0.509 | 0.512 | 0.503 |
| MultiRC | 0.516 | 0.533 | 0.522 | 0.507 | 0.506 | 0.527 |
| Average Performance | 0.442 | 0.459 | 0.464 | 0.465 | 0.450 | 0.456 |
| *1B Model Pretrained on 20B Tokens* | | | | | | |
| Social IQA | 0.387 | 0.411 | 0.406 | 0.406 | 0.394 | 0.401 |
| HellaSwag | 0.375 | 0.427 | 0.431 | 0.436 | 0.410 | 0.410 |
| PiQA | 0.658 | 0.684 | 0.693 | 0.691 | 0.684 | 0.680 |
| OpenBookQA | 0.278 | 0.298 | 0.300 | 0.304 | 0.299 | 0.302 |
| Lambada | 0.301 | 0.359 | 0.348 | 0.353 | 0.334 | 0.339 |
| SciQ | 0.802 | 0.822 | 0.809 | 0.828 | 0.821 | 0.818 |
| ARC Easy | 0.482 | 0.508 | 0.512 | 0.518 | 0.500 | 0.499 |
| COPA | 0.683 | 0.692 | 0.713 | 0.708 | 0.678 | 0.698 |
| RACE | 0.306 | 0.319 | 0.313 | 0.314 | 0.305 | 0.300 |
| LogiQA | 0.259 | 0.258 | 0.269 | 0.272 | 0.268 | 0.267 |
| WinoGrande | 0.513 | 0.527 | 0.541 | 0.512 | 0.535 | 0.533 |
| MultiRC | 0.523 | 0.504 | 0.510 | 0.530 | 0.529 | 0.548 |
| Average Performance | 0.464 | 0.484 | 0.487 | 0.489 | 0.480 | 0.483 |
| Estimated FLOPs | 0 | $3.7 \times 10^{19}$ (100%) | 0 | $3.5 \times 10^{18}$ (9.46%) | $9.66 \times 10^{16}$ (0.26%) | $9.66 \times 10^{16}$ (0.26%) |

### 4.2.1 EXPERIMENTAL SETUP

**Dataset & Baseline** The Pile dataset (Gao et al., 2021) is an 825 GB English text corpus for the pretraining of large language models. Following RegMix (Liu et al., 2024), we also just use the 17 components of The Pile that do not have copyright issues. And we should find the data mixture to achieve lower validation loss on Pile-CC for better downstream task performance. We also compare our approach with various baselines, such as Human (Based on the data size), DoReMi (Xie et al., 2023a), and RegMix (Liu et al., 2024). Pile-CC Only (Just train the model on the Pile-CC sub dataset) is designed for verifing that there is a strong correlation between Pile-CC's validation loss and downstream performance. The data mixture of different baselines are shown in Table 4.

**Training Setup** We pretrained various sizes of Transformer (Vaswani et al., 2017) Decoder-only models from scratch using a next-token prediction loss. The model parameters range from 106M to 1B. Following (Hoffmann et al., 2022b), the computed token number for the different models is 20 times the parameter number of current model. All the models have a batch size of 1M tokens, and the maximum sequence length is $4096$. We use the Adam Kingma & Ba (2017) optimizer with gradient clip of $1.0$. The learning rate linearly warms up to a maximum learning rate of 6e-4 over the first $1,000$ steps, then decreases to 0 using a cosine learning rate scheduler at the end of training stage. The detailed parameters of models we used are shown in the Table 6.

**Evaluation** First, we observed the performance on Pile-CC's validation loss on different model sizes as shown in Figure 5. Then we evaluated the performance of different data mixture using following benchmarks: Social IQA (Sap et al., 2019), HellaSwag (Zellers et al., 2019), PiQA (Bisk et al., 2019), OpenBookQA (Mihaylov et al., 2018), Lambada (Paperno et al., 2016), SciQ (Welbl et al., 2017), ARC Easy (Clark et al., 2018), COPA (Gordon et al., 2012), RACE (Lai et al., 2017), LogiQA (Liu et al., 2021), WinoGrande (Sakaguchi et al., 2021), and MultiRC (Khashabi et al., 2018). We utilize LM Evaluation Harness (Gao et al., 2024) to evaluate these models and report the average score across 0-shot to 5-shot settings in Table 3.

### 4.2.2 Experimental Results

First, we sample $1,000$ samples from each component from The Pile and Pile-CC validation set and use the Meta-Domain Classifier to calculate the domain vector of each dataset. We generate $100,000$ different data mixture from a Dirichlet distribution based on the token distribution. Using these mixtures, we predict the optimal data mixture by applying Equation 5 (DOMAIN2VEC+DA$^2$) and Equation 7 (DOMAIN2VEC+RegMix). To avoid the over-fitting of language models, each subset of The Pile is trained for at most one epoch. We also apply rejection sampling to eliminate all the unreasonable data mixtures. As a result, the optimal data mixture predicted by DOMAIN2VEC may vary depending on the size of language models.

As illustrated in Figure 5, *our proposed* **DOMAIN2VEC + DA$^2$** *and* **DOMAIN2VEC + REGMIX** *could significantly improve the training efficiency on Pile-CC compared to Human* (Using the original data mixture from The Pile). Specifically, DOMAIN2VEC + DA$^2$ and DOMAIN2VEC + REGMIX require only about $55.38\%$ and $51.50\%$ of the training steps, respectively, to reach the same validation loss as Human. Furthermore, under equivalent compute budget, DOMAIN2VEC + DA$^2$ and DOMAIN2VEC + REGMIX reduce the validation loss by approximately $4.04\%$ and $4.64\%$, respectively, compared to Human. In Table 3, we report the performance of models trained on data mixtures derived from various baselines across a wide range of downstream tasks. It can be observed that the Pile-CC Only shows an average accuracy improvement of $4.27\%$ over Human, indicating that training on more tokens from Pile-CC does enhance the downstream task performance of language models. More importantly, *our proposed* **DOMAIN2VEC + DA$^2$** *and* **DOMAIN2VEC + REGMIX,** *utilizing only about* $0.26\%$ *of the FLOPs required by DoReMi, could identify data mixtures that achieve performance comparable to DoReMi, RegMix and Pile-CC Only*. This demonstrates both the effectiveness and computational

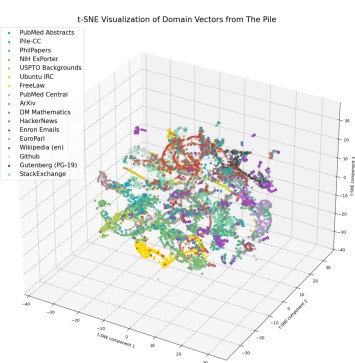

Figure 6: We use t-SNE to visualize domain vectors from different sub-datasets from The Pile. This figure indicates that different sub-datasets may contain data from the same Meta-Domain, which explains why different datasets can mutually benefit the training of each other.

efficiency of DOMAIN2VEC. At the same time, it is important to note that while achieving comparable performance, our method does not allocate an excessively high proportion on Pile-CC training dataset as DoReMi and RegMix do. This suggests that different datasets might mutually benefit the training of other datasets. To investigate the cause of this phenomenon, we used t-SNE (Van der Maaten & Hinton, 2008) to visualize the domain vector of each component of The Pile, as shown in Figure 6. This figure reveals that different datasets can contain data belonging to same meta-domain, and datasets like Pile-CC, Wikipedia, PhilPapers encompass data from many different meta-domains. This overlap between datasets suggests that the domain vector effectively captures the underlying features of different datasets, explai

## 5 Related Work

Recently, there has been a amount of research focusing on finding the optimal data mixture, which could be broadly categorized into two lines. The first line **implicitly adjusts** the data mixture by down-sampling data from various datasets via finding high-quality data. For instance, Lin et al. (2024) propose RHO-1, which leverages Selective Language Models to select useful tokens to align the data mixture with the ideal ratio. Rather than selecting high-quality data at the token level, Ankner et al. (2024) utilize the perplexity of small reference models to filter out low-quality samples.

Additionally, Thakkar et al. (2023) demonstrate that the Influence Score could guide the process of data re-weighting. After that, Thakkar et al. (2023) propose an online data selection method that eliminates the need of any reference model. The second line of research emphasizes modeling the relationship between data mixture and the performance of language models, which **explicitly adjusts** the data mixture across different datasets. The most straightforward approach is to observe the performance of various data mixtures and then select the optimal one, as demonstrated during the training of Gopher (Rae et al., 2022). However, this approach comes with high training costs, making it challenging to scale for larger models. To address this issue, Xie et al. (2023a) propose DoReMi, which utilizes a small proxy model to re-weight data from different domains, improving the training efficiency of larger models to some extent. However, DoReMi still requires a well-trained reference model beforehand, which introduce additional computational costs, and it is challenging to define what constitutes an ideal reference model. In response, Fan et al. (2023) introduce DoGE, a method that uses a min-max optimization to train a proxy model for obtaining better domain weights. This approach assigns larger weights to domains that either contribute to learning in other domains or are themselves more challenging to learn. Chen et al. (2023) also propose a skills-based framework to dynamically adjust data mixtures during model training. While the aforementioned methods consider the relationship between data mixture and the performance of language models, they typically provide a single data mixture rather than modeling a functional relationship. Inspired by the scaling law (Kaplan et al., 2020; Hoffmann et al., 2022b), Ye et al. (2024) propose Data Mixing Laws, which describes this relationship using an exponential form. Similarly, Ge et al. (2024) introduce BiMix, a scaling law that accounts for both compute consumption and the data mixture. Both Que et al. (2024) and Gu et al. (2024) develop scaling laws for continued pretrain, considering the data mixtures between pretrained and continued pretrained datasets. Notably, Gu et al. (2024) accurately predict that the pretrained domain loss would first increase and then decrease during continued pre-training, and introduce critical mixture ratios to mitigate catastrophic forgetting in the pretrained domain. More recently, Liu et al. (2024) propose using a Linear Regression Model to fit the validation loss of different data mixtures, demonstrating a strong correlation.

While prior works have shown promising results, they have some issues as follows: **1) Computational Efficiency**: For instance, the estimated FLOPs of DoReMi and RegMix is $3.7 \times 10^{19}$ and $3.5 \times 10^{18}$. **2) Lack of Scalability**: When the components of the training dataset change (i.e., add some new datasets), the previously fitted functions like Ye et al. (2024) and Liu et al. (2024) cannot be directly applied to the updated scenario. This limitation arises because the dimension of the independent variable (i.e., the number of different datasets) in these fitted relationships is fixed. As a result, we need to resample different data mixtures, then retrain some proxy models, and perform the fitting again. In this paper, we propose a novel concept DOMAIN2VEC, which decomposes any dataset into a linear combination of several Meta-Domains to capture underlying features of datasets. DOMAIN2VEC shares some ideas with some prior works in the field of Meta-Learning, such as Jomaa et al. (2021) and Chen et al. (2024). These works have explored dataset representation in latent spaces. While sharing the concept of latent space representation for datasets, DOMAIN2VEC differs in both purpose and implementation and we focus on language model pretraining data mixture. Then we propose **D**ISTRIBUTION **A**LIGNMENT **A**SSUMPTION, a training-free manner to identify the optimal data mixture for language model pretraining. Importantly, using DOMAIN2VEC tools we provided, all fitting experiments are conducted in the dimension of Meta-Domains. When training datasets change, we can still map them as linear combinations of several Meta-Domains, which greatly enhance the scalability of prior works (Xie et al., 2023a; Ye et al., 2024; Liu et al., 2024).

## 6 CONCLUSIONS

In this work, we introduce DOMAIN2VEC, a novel concept to capture the underlying features of datasets by decomposing datasets into a linear combination of several "Meta-Domains". We also propose an efficient method to acquire vectorized representation (domain vector) for any given dataset. Based on the domain vector, we introduce a training-free approach by *Distribution Alignment Assumption* ($DA^2$) for language models pretraining to find the optimal data mixture. By leveraging DOMAIN2VEC, we greatly enhance the scalability of previous methods without re-training as training datasets change. Experimental results show that DOMAIN2VEC could use less computation costs to find the data mixture with better text generation ability and downstream task performance. DOMAIN2VEC could serve as a strong and efficient baseline, and we hope that this work will provide some insights into the data mixture optimization for language models pretraining.

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

# A APPENDIX

## A.1 DETAILED DESCRIPTION OF THE DISTRIBUTION ALIGNMENT ASSUMPTION

In this section, we will introduce the detailed description of the Distribution Alignment Assumption for language model pretaining.

In the scenario of finding the optimal data mixture for language model pretraining, the validation set $\mathcal{D}_{valid}$ is fixed, and we should adjust the data mixture to construct the training set $\mathcal{D}_{train}$ to achieve lower validation loss calculated by Equation 9, where $\hat{\theta}$ is parameters of a pretrained language model.

$$\mathbb{E}_{X \sim \mathcal{D}_{valid}} - \log P(X|\hat{\theta}) = \mathbb{E}_{X \sim \mathcal{D}_{valid}} \sum_{i=1}^{|X|} - \log(P(x_i|x_{<i}, \hat{\theta})) \tag{9}$$

Typically, we pretrain language models via next token prediction (Radford, 2018) like Equation 10.

$$\hat{\theta} = \arg\max_{\theta} \mathbb{E}_{X \sim \mathcal{D}_{train}} \log P(X|\theta)$$

$$= \arg\max_{\theta} \mathbb{E}_{X \sim \mathcal{D}_{train}} \sum_{i=1}^{|X|} \log(P(x_i|x_{<i}, \theta)) \tag{10}$$

That is, we need to find a $\hat{\theta}$ that maximizes the expected probability of $X \sim \mathcal{D}_{train}$, which is also known as Maximum Likelihood Estimation (MLE). When the data distributions of $\mathcal{D}_{\text{train}}$ and $\mathcal{D}_{\text{valid}}$ are aligned, the optimization target of language models pretraining process equals find a $\hat{\theta}$ that maximizes the expected probability of $X \sim \mathcal{D}_{valid}$. Therefore, we introduce the Distribution Alignment Assumption for language model pretaining, a novel method to find the optimal data mixture without training. After that, we propose to use the Meta-Domain Classifier to capture some underlying features of datasets which could helps modeling the data distribution of different datasets.

## A.2 ALGORITHM

In Algorithm 1, we show the pseudo code for acquiring the domain vector for pretraining datasets.

In Algorithm 2 and 3, we show the pseudo code for how to use DOMAIN2VEC to find the optimal data mixture, including Distribution Alignment Assumption, and applying DOMAIN2VEC to RegMix (Liu et al., 2024).

---

**Algorithm 1** DOMAIN2VEC

**Require:** Training Datasets $\mathcal{D}_{train} = \{\mathcal{D}_1, \mathcal{D}_2, ..., \mathcal{D}_k\}$ , Validation Dataset $\mathcal{D}_{valid}$, Meta-Domain Classifier Classifier

1:
2: Domain Vectors $\boldsymbol{V}_{train} = []$
3: **for** $i = 1$ to $k$ **do**
4:     Sample $N$ data points from $\mathcal{D}_i$
5:     $\boldsymbol{v}_i = \frac{1}{N} \sum_{j=0}^{N-1} \text{Classifier}(text_j)$, where $text_j \in \mathcal{D}_i$     ▷ Get Domain Vectors of $\mathcal{D}_{train}$
6:     $\boldsymbol{V}_{train} = [\boldsymbol{V}_{train}, \boldsymbol{v}_i]$
7: **end for**
8:
9: Sample $N$ data points from $\mathcal{D}_{valid}$
10: $\boldsymbol{q}_{valid} = \frac{1}{N} \sum_{j=0}^{N-1} \text{Classifier}(text_j)$, where $text_j \in \mathcal{D}_{valid}$     ▷ Get Domain Vector of $\mathcal{D}_{valid}$
11:
12: **return** $\boldsymbol{V}_{train} = [\boldsymbol{v}_1, \boldsymbol{v}_2, ..., \boldsymbol{v}_k]$, $\boldsymbol{q}_{valid}$

---

## A.3 DATA MIXTURE OF DIFFERENT METHODS

In this section, we will show the data mixture on The Pile (Gao et al., 2021) of different methods we used in this paper for reproduction. In Table 4, we show the optimal data mixture predicted by

---

**Algorithm 2** DOMAIN2VEC+DA$^2$

---

**Require:** Domain Vectors of Training Datasets $V_{train} = [v_1, v_2, ..., v_k]$, Domain Vectors of Validation Dataset $q_{valid}$, Token Distribution of Training Datasets $a_{train} = [\alpha_1, \alpha_2, ..., \alpha_k]$.

 1:
 2: Sample $K$ candidates data mixture $r_i$ from $\text{Dirichlet}(a_{train})$
 3:
 4: The Optimal Data Mixture $r^* = r_1$        ▷ Initialize the optimal data mixture
 5:
 6: **for** $i = 2$ to $k$ **do**
 7:    **if** $\text{Dist}(V_{train} \cdot r, q_{valid}) < \text{Dist}(V_{train} \cdot r^*, q_{valid})$ **then**    ▷ Updata the optimal data mixture
 8:      $r^* = r_i$
 9:    **end if**
10: **end for**
11:
12: **return** the optimal data mixture $r^*$

---

**Algorithm 3** DOMAIN2VEC+RegMix

---

**Require:** Domain Vectors of Training Datasets $V_{train} = [v_1, v_2, ..., v_k]$, Domain Vectors of Validation Dataset $q_{valid}$, Token Distribution of Training Datasets $a_{train} = [\alpha_1, \alpha_2, ..., \alpha_k]$, Fitted Linear Regression Model for Each Meta-Domain $\mathcal{L}^{\mathcal{D}_i^*}(p)$.

 1:
 2: Sample $K$ candidates data mixture $r_i$ from $\text{Dirichlet}(a_{train})$
 3:
 4: The Optimal Data Mixture $r_* = r_1$        ▷ Initialize the optimal data mixture
 5: $\mathcal{L}(r^*) = \sum_{i=0}^{n-1} q_i \cdot \mathcal{L}^{\mathcal{D}_i^*}(V_{train} \cdot r_1)$
 6:
 7: **for** $i = 2$ to $k$ **do**
 8:    **if** $\mathcal{L}(r_i) < \mathcal{L}(r^*)$ **then**        ▷ Updata the optimal data mixture
 9:      $r^* = r_i$
10:      $\mathcal{L}(r^*) = \mathcal{L}(r_i)$
11:    **end if**
12: **end for**
13:
14: **return** the optimal data mixture $r^*$

---

DOMAIN2VEC + DA$^2$ and DOMAIN2VEC + RegMix. It should be noted that, to avoid the over-fitting problem, any subset of The Pile (Gao et al., 2021) will be only trained at most one epoch. Because we adopt rejection sampling to filter out certain unreasonable data mixtures. The data mixture predicted may change as model sizes change.

A.4 EXPERIMENTAL RESULTS OF PILOT STUDY

In this section, we report the validation loss on various datasets Arxiv, C4, Book3, PG19 from Red-Pajama (Computer, 2023), and BookCorpus2, DM Mathematics, Enron Emails, FreeLaw, Hack-erNews, NIH ExPorter, OpenSubtitles, OpenWebText2, PhilPapers, PubMed Abstracts, PubMed Central, USPTO Backgrounds, Ubuntu IRC, Youtube Subtitles from The Pile (Gao et al., 2021) in Figure 4, Figure 9 and Figure 8. According to the experimental results, we find that 1) *for different validation sets, the ranking of mixture ratios varies significantly*. 2) *for the same validation set, the data mixture ranking of validation loss on identical validation dataset does not change with the variation in model parameters*. We hope our experimental results and findings could provide some insights to the community about efficiently finding the optimal data mixture.

Table 4: The data mixture of The Pile (Gao et al., 2021) from different baselines, which aligns with the data mixture used in Liu et al. (2024).

| Data Mixture | Human | DoReMi | Pile-CC Only | RegMix |
|---|---|---|---|---|
| ArXiv | 0.134 | 0.004 | 0.0 | 0.001 |
| FreeLaw | 0.049 | 0.005 | 0.0 | 0.001 |
| NIH ExPorter | 0.007 | 0.008 | 0.0 | 0.001 |
| PubMed Central | 0.136 | 0.006 | 0.0 | 0.003 |
| Wikipedia (en) | 0.117 | 0.086 | 0.0 | 0.016 |
| DM Mathematics | 0.025 | 0.002 | 0.0 | 0.0 |
| Github | 0.054 | 0.022 | 0.0 | 0.0 |
| PhilPapers | 0.003 | 0.034 | 0.0 | 0.0 |
| Stack Exchange | 0.118 | 0.019 | 0.0 | 0.0 |
| Enron Emails | 0.004 | 0.009 | 0.0 | 0.002 |
| Gutenberg (PG-19) | 0.025 | 0.009 | 0.0 | 0.002 |
| Pile-CC | 0.142 | 0.743 | 1.0 | 0.87 |
| Ubuntu IRC | 0.009 | 0.011 | 0.0 | 0.064 |
| EuroParl | 0.005 | 0.008 | 0.0 | 0.0 |
| HackerNews | 0.01 | 0.016 | 0.0 | 0.012 |
| PubMed Abstracts | 0.107 | 0.014 | 0.0 | 0.024 |
| USPTO Backgrounds | 0.053 | 0.004 | 0.0 | 0.002 |

Table 5: The optimal data mixture predicted by DOMAIN2VEC + DA$^2$ and DOMAIN2VEC + Reg-Mix. To avoid the over-fitting problem, any subset of The Pile (Gao et al., 2021) will be trained at most one epoch. And we adopt rejection sampling to filter out certain unreasonable data mixtures. Thus, the data mixture predicted may change as model sizes change.

| Data Mixture | DOMAIN2VEC+DA$^2$ | | | | DOMAIN2VEC+RegMix | | | |
|---|---|---|---|---|---|---|---|---|
| | 106M | 290M | 595M | 1B | 106M | 290M | 595M | 1B |
| ArXiv | 0.0131 | 0.0131 | 0.0389 | 0.0431 | 0.0152 | 0.0070 | 0.0114 | 0.0103 |
| FreeLaw | 0.0076 | 0.0076 | 0.0316 | 0.0305 | 0.0395 | 0.0267 | 0.0339 | 0.0268 |
| NIH ExPorter | 0.0008 | 0.0008 | 0.0028 | 0.0023 | 0.0000 | 0.0199 | 0.0000 | 0.0000 |
| PubMed Central | 0.0773 | 0.0773 | 0.0519 | 0.0704 | 0.0343 | 0.0576 | 0.0099 | 0.0518 |
| Wikipedia (en) | 0.2970 | 0.2970 | 0.2049 | 0.2126 | 0.0847 | 0.0101 | 0.1014 | 0.2577 |
| DM Mathematics | 0.0003 | 0.0003 | 0.0056 | 0.0026 | 0.0177 | 0.0018 | 0.0011 | 0.0008 |
| Github | 0.0096 | 0.0096 | 0.0290 | 0.0298 | 0.0034 | 0.0538 | 0.0500 | 0.0138 |
| PhilPapers | 0.0018 | 0.0018 | 0.0093 | 0.0025 | 0.0118 | 0.0005 | 0.0333 | 0.0401 |
| Stack Exchange | 0.0464 | 0.0464 | 0.0661 | 0.0585 | 0.0698 | 0.0430 | 0.1199 | 0.0262 |
| Enron Emails | 0.0000 | 0.0000 | 0.0009 | 0.0000 | 0.0018 | 0.0000 | 0.0000 | 0.0000 |
| Gutenberg (PG-19) | 0.0217 | 0.0217 | 0.0484 | 0.0370 | 0.0467 | 0.0223 | 0.0007 | 0.0252 |
| Pile-CC | 0.4338 | 0.4338 | 0.3191 | 0.3814 | 0.5370 | 0.6323 | 0.5546 | 0.4704 |
| Ubuntu IRC | 0.0022 | 0.0022 | 0.0063 | 0.0072 | 0.1019 | 0.0123 | 0.0161 | 0.0069 |
| EuroParl | 0.0003 | 0.0003 | 0.0042 | 0.0040 | 0.0070 | 0.0037 | 0.0116 | 0.0000 |
| HackerNews | 0.0154 | 0.0154 | 0.0521 | 0.0199 | 0.0028 | 0.0551 | 0.0170 | 0.0673 |
| PubMed Abstracts | 0.0596 | 0.0596 | 0.0739 | 0.0532 | 0.0259 | 0.0102 | 0.0190 | 0.0017 |
| USPTO Backgrounds | 0.0130 | 0.0130 | 0.0549 | 0.0449 | 0.0004 | 0.0438 | 0.0201 | 0.0010 |

Table 6: The parameters of different models we used in Section 4.1 and Section 4.2. When calculating the model parameters, we do not take into account the embedding layer and the language model head layer.

| Parameter | Text Generation | | Downstream Task | | | |
|---|---|---|---|---|---|---|
| | 83M | 1.6B | 106M | 290M | 595M | 1B |
| Hidden Size | 768 | 2048 | 768 | 1280 | 1536 | 2048 |
| FFN Hidden Size | 2048 | 5504 | 2048 | 3392 | 4096 | 5440 |
| Num of Layers | 12 | 24 | 15 | 15 | 21 | 21 |
| Num of Heads | 12 | 16 | 12 | 10 | 12 | 32 |
| Max Seq Length | 4096 | 4096 | 4096 | 4096 | 4096 | 4096 |
| Vocab Size | 128256 | 128256 | 151936 | 151936 | 151936 | 151936 |
| RoPE Base | 10000 | 10000 | 10000 | 10000 | 10000 | 10000 |

Table 7: Downstream Task Performance of different data mixture on 106M Model. Similiar to Liu et al. (2024), Human refers the original data mixture from The Pile. Pile-CC Only refers only training on the Pile-CC subset. The data mixture and estimated flops of DoReMi and RegMix are from Liu et al. (2024).

| Benchmark | Human | DoReMi | Pile-CC Only | RegMix | DOMAIN2VEC + DA$^2$ | DOMAIN2VEC + RegMix |
|---|---|---|---|---|---|---|
| | | | *106M Model Pretrained on 2B Tokens* | | | |
| Social IQA | 0.340 | 0.349 | 0.353 | 0.356 | 0.339 | 0.342 |
| HellaSwag | 0.268 | 0.268 | 0.269 | 0.269 | 0.267 | 0.264 |
| PiQA | 0.573 | 0.584 | 0.580 | 0.586 | 0.579 | 0.583 |
| OpenBookQA | 0.245 | 0.251 | 0.249 | 0.242 | 0.245 | 0.249 |
| Lambada | 0.065 | 0.099 | 0.102 | 0.091 | 0.091 | 0.090 |
| SciQ | 0.550 | 0.520 | 0.509 | 0.537 | 0.549 | 0.518 |
| ARC Easy | 0.329 | 0.339 | 0.335 | 0.337 | 0.334 | 0.331 |
| COPA | 0.525 | 0.570 | 0.572 | 0.585 | 0.578 | 0.557 |
| RACE | 0.236 | 0.254 | 0.246 | 0.251 | 0.240 | 0.244 |
| LogiQA | 0.282 | 0.280 | 0.271 | 0.274 | 0.268 | 0.286 |
| WinoGrande | 0.516 | 0.516 | 0.502 | 0.508 | 0.506 | 0.499 |
| MultiRC | 0.539 | 0.520 | 0.515 | 0.533 | 0.541 | 0.544 |
| Average Performance | 0.372 | 0.379 | 0.375 | 0.381 | 0.378 | 0.376 |
| Estimated FLOPs | 0 | $3.7 \times 10^{19}$ (100%) | 0 | $3.5 \times 10^{18}$ (9.46%) | $9.66 \times 10^{16}$ (0.26%) | $9.66 \times 10^{16}$ (0.26%) |

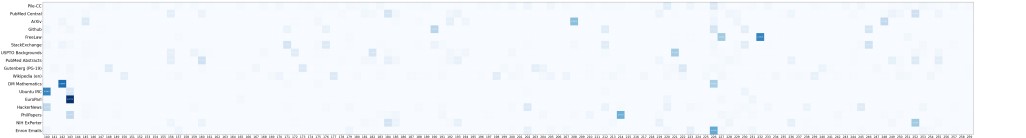

Figure 7: The Domain Vector of each sub-dataset of The Pile (Gao et al., 2021), where each row corresponds to a sub-dataset and each column corresponds to a Meta-Domain. The higher the proportion of data belonging to a particular Meta-Domain, the closer the color of the corresponding cell is to blue). Additionally, since The Pile primarily consists of English texts, we only display the distribution on English Meta-Domains for clarity.

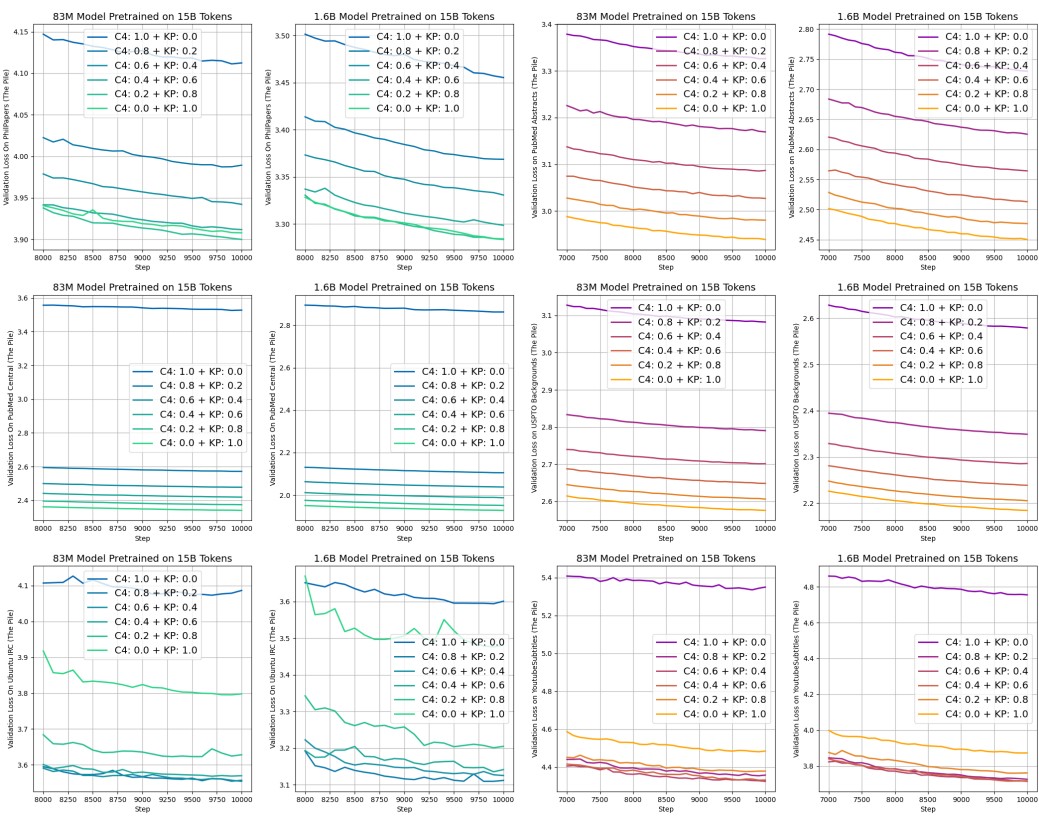

Figure 8: The validation loss on different dataset of models trained using data mixture in Table 1.

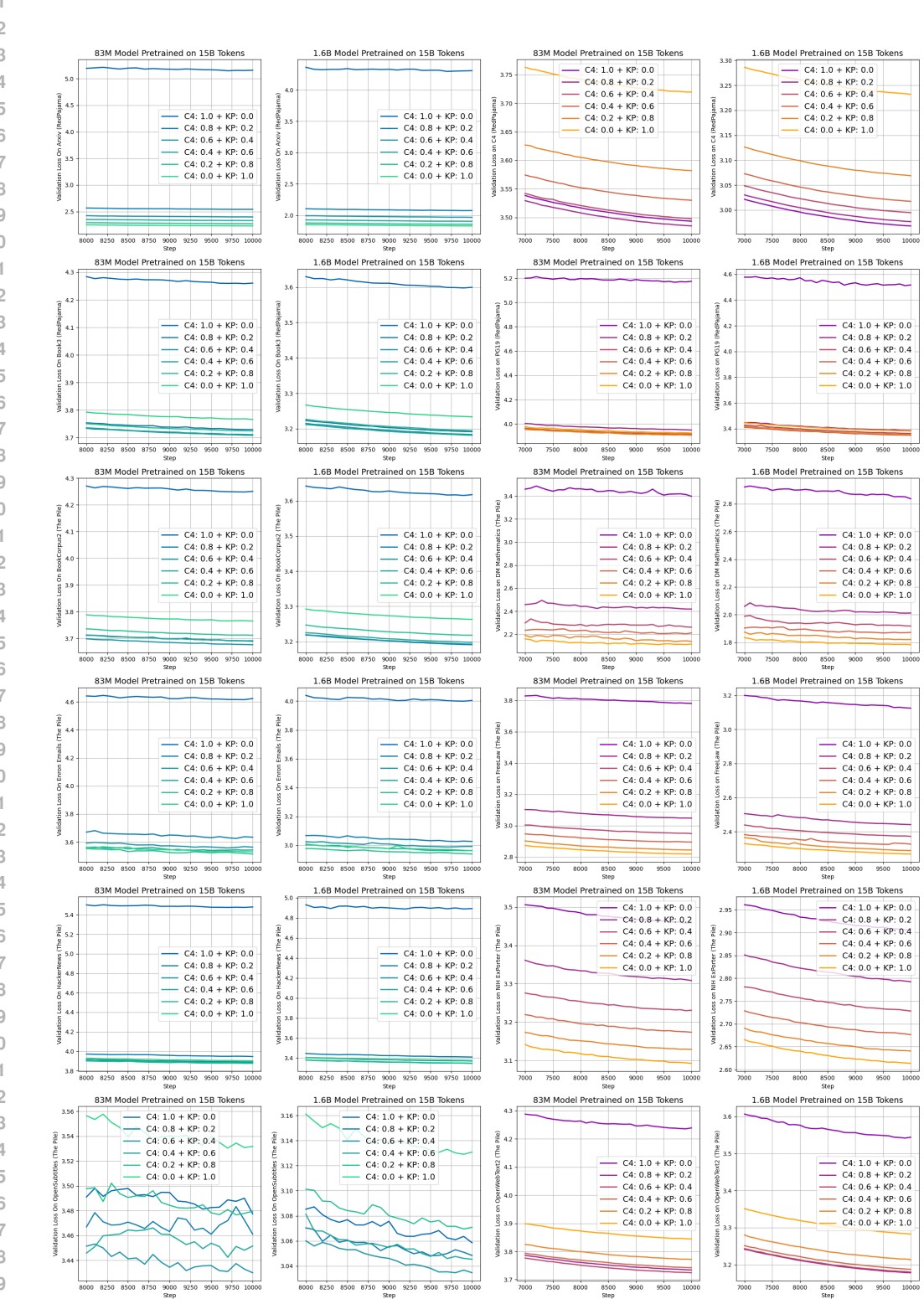

Figure 9: The validation loss on different dataset of models trained using data mixture in Table 1.

