# OpenReview forum: "Domain2Vec: Vectorizing Datasets to Find the Optimal Data Mixture without Training"
_ICLR.cc/2025/Conference — Submitted to ICLR 2025_

### Official Review · Reviewer_tRxw · 2024-10-24

**Soundness:** 3
**Presentation:** 2
**Contribution:** 2
**Rating:** 6
**Confidence:** 3

**Summary:**

This paper introduces Domain2Vec which decomposes data sets into meta-domain clusters from which data mixture ratios can be easily computed. The paper then uses their proposed strategy, DA2, to select better data mixtures than DoReMi.

**Strengths:**

The methodology is intuitive and easy to follow. The results demonstrate that the proposed method is effective.

**Weaknesses:**

The constructed meta-domains are heavily biased towards the initial data sources. This work collates 260 meta-domains over 100 sources containing English, Chinese, and Code. A practitioner in a distinctly different domain (e.g., German) may not be able to use the constructed domain2vec, and would have to first create their own domain2vec setup. This poses a type of chicken-and-egg problem on the usefulness of domain2vec.

**Questions:**

* How do we construct the vocabulary of meta-domains given a set of sources?
* How do we choose N, i.e., the number of points needed to get a confident probability vector?
* When training the meta-domain classifier, is an arbitrary text belonging to a single meta-domain or can the arbitrary text also belong to multiple meta-domains? Is the meta-domain classifier a multi-class or multi-label problem? How do you handle precision/recall over long-tail meta-domains?
* Rather than fitting a meta-domain classifier, a simple baseline would be to leverage learned feature embeddings to automatically cluster the data into a set of meta-domains. How well would this perform versus the meta-domain vocabulary and the classifier?
* Ultimately, DA2 is trying to match two categorical probability vectors. Why use a Huber loss rather than, e.g., distributional measures (divergence, Wasserstein distance, etc.)?

---

> ### Author Response · Authors · 2024-11-22
> **Response to Reviewer tRxw [1/2]**
>
> Dear Reviewer tRxw,
>
> Thanks for your insightful review! We will respond to your questions one by one.
>
> ## Q1: How do we construct the vocabulary of meta-domains given a set of sources?
>
> In the "Construct the Vocabulary of Domain2Vec" paragraph in Section 2 (lines 127-146), we explain how we construct several Meta-Domains from our collected pretraining datasets. Specifically, we used K-means to cluster on our collected text data of over 5TB size, where texts belonging to the same cluster form a Meta-Domain. Finally, we obtained 260 Meta-Domains. We use the term, "vocabulary" to denote a set composed of Meta-Domains, as an analogy of Word2Vec.
>
> ## Q2: How do we choose N, i.e., the number of points needed to get a confident probability vector？
>
> Theoretically, a larger N would be better. In this paper, we sampled 1,000 points from each validation set to calculate the domain vector. Our experimental results show that 1,000 points are sufficient for good performance. In practice, one can determine the sample size by adding samples until the calculated domain vector becomes very stable.
>
> ## Q3: When training the meta-domain classifier, is an arbitrary text belonging to a single meta-domain or can the arbitrary text also belong to multiple meta-domains? Is the meta-domain classifier a multi-class or multi-label problem? How do you handle precision/recall over long-tail meta-domains?
>
> 1. During the training phase, we train the meta-domain classifier with a **single-label** problem. The classifier predicts which meta-domain a text belongs to. During inference time, the meta-domain classifier outputs a probability distribution over meta-domains, rather than a one-hot vector.
> 2. When training the Meta-Domain classifier, we use the same sample size for different Meta-Domains, which helps mitigate possible class imbalance problem. Moreover, **there is no real long-tail meta domains**. even the smallest Meta-Domain contains approximately 2GB texts as shown in the table below.
>
> | Disk Size (GB) | Number of Meta-Domains |
> | -------------- | ---------------------- |
> | 1.93 - 5.32    | 13                     |
> | 5.32 - 8.71    | 18                     |
> | 8.71 - 12.10   | 25                     |
> | ...            | ...                    |
>
>
> ## Q4: Rather than fitting a meta-domain classifier, a simple baseline would be to leverage learned feature embeddings to automatically cluster the data into a set of meta-domains. How well would this perform versus the meta-domain vocabulary and the classifier?
>
> Great question! Technically, your mentioned baseline is totally feasible. However, we show the reasons for our practice from the following aspects:
>
> 1. The Embedding models like BGE have a context window limited to 512 tokens, while pre-training data is typically longer than 512 tokens. In contrast, our Meta-Domain Classifier, trained based on Qwen, can handle context lengths of up to 8k tokens or even longer.
>
> 2. More importantly, our Meta-Domain classifier can output very specific probability distribution on the meta-domains while the baseline can only output a hard assignment (0 or 1). There are indeed some $k$-Nearest Neighbors algorithms that can output soft scores, but the score is indirectly based on the distance to the center of clusters. The efficiency of $k$-NN is also limited because it is a lazy learning algorithm and puts time complexity into inference time.
>
> ## Q5: Ultimately, $DA^2$ is trying to match two categorical probability vectors. Why use a Huber loss rather than, e.g., distributional measures (divergence, Wasserstein distance, etc.)?
>
> Another great question! Technically, Huber loss combines the advantages of L1 and L2 distance. We add the results of different distributional measures based on Table 2 in our paper. As shown in the table, Huber Loss shows better performance than L1/L2/JS Distance.
>
> | Distributional Measure | Pearson | Spearman |
> | ---------------------- | ------- | -------- |
> | Huber Loss             | 0.58330 | 0.66570  |
> | JS Distance            | 0.45267 | 0.50000  |
> | L1 Distance            | 0.48303 | 0.54000  |
> | L2 Distance            | 0.57198 | 0.64286  |
>
> Additionally, Wasserstein distance is a very great option, to be honest. However, it would require an extra metric space matrix, $\boldsymbol{M}$, to measure the distance between two vectors. In this work, the metric space $\boldsymbol{M} (260 \times 260)$ is actually the "dataset transition cost" between each two Meta-Domains, and is non-trivial. Each element in $\boldsymbol{M}$, $c_{ij}$ could be estimated via $L_{ij}$, the loss at Meta-Domain $j$ after training Meta-Domain $i$, which requires additional computational resources. Considering that Huber Loss already achieved very positive results, we did not conduct this experiment. We believe that Wasserstein distance can also present a positive result (even better) if the metric space is well estimated, and we leave this for future work.

---

> > ### Author Response · Authors · 2024-11-22
> > **Response to Reviewer tRxw [2/2]**
> >
> > ## W1: The constructed meta-domains are heavily biased towards the initial data sources. A practitioner in a distinctly different domain (e.g., German) may not be able to use the constructed domain2vec, and would have to first create their own domain2vec setup.
> >
> > We answer the question from the two following aspects
> >
> > 1. As mentioned in Line 130, we suggest that "The original data for constructing Meta-Domains should be as diverse and large as possible". And in this paper, we provide a practical implementation across three language - English, Chinese, and code - to demonstrate the feasibility of Domain2Vec. We consider this to be the core contribution of this paper.
> >
> > 2. At this stage of the study, for other language, it indeed requires its own Domain2Vec setup. However, we want to emphasize that this process is still a one-time effort. Once a Domain2Vec (e.g. German) model is made and ready, all German datasets can be input into this model to determine the optimal mixing ratio.
> >
> > 3. From another perspective, we fully agree that multilingualism is crucial to LLMs. We are also working on to include more languages.  It is quite possible that future researchers could develop a multilingual Domain2Vec encompassing almost all languages in the world. At that time, creating a separate Domain2Vec setup would no longer be necessary. This paper does not aim to cover all languages but demonstrates the feasibility of computing the optimal mixing ratio using just a simple domain vector.
> >
> > We sincerely appreciate the time and effort you've put into reviewing our work. We have responded to each of your questions in detail and would be happy to provide any additional information you might need.
> >
> > Best regard,
> >
> > Authors of Paper 9648

---

> > > ### Comment · Reviewer_tRxw · 2024-11-22
> > > **Thanks for the detailed rebuttal**
> > >
> > > Thanks for your comments, which have cleared up some of my confusions. I have increased my score accordingly.
> > >
> > > The clarification on choice of loss is useful --- both that Huber loss empirically demonstrates successful results and that the distribution matching approach may invite many design decisions. I still believe that a distribution matching approach (whether Wasserstein or any other appropriate distribution-based metric) is the intuitive approach and I encourage you to explore it (in future work if needed).
> > >
> > > I still feel that simple baselines (like clustering) are useful, especially as it permits automatic learning of a meta domain vocabulary. I agree with the token argument and I hope that your method should outperform the baseline.

---

> > > > ### Author Response · Authors · 2024-11-24
> > > > **Response to Reviewer tRxw**
> > > >
> > > > Dear Reviewer tRxw,
> > > >
> > > > We greatly appreciate your review and are pleased to have addressed your primary concerns.
> > > >
> > > > We agree that the distribution matching approach is a worthwhile direction for future work.
> > > >
> > > > Additionally, we have added an embedding-based baseline, denoted as KNN.
> > > >
> > > >     First, for the training and validation datasets in section 4.1, we sampled 1000 examples from each dataset.
> > > >
> > > >     Then, we used bge-small-v1.5 (since the datasets in section 4.1 are in English) to obtain embeddings for samples from each dataset and used mean pooling to get unique embeddings for each dataset.
> > > >
> > > >     Meanwhile, we also used bge-small-v1.5 to obtain embeddings for the data in each Meta-Domain.
> > > >
> > > >     Then, we used KNN (based on Euclidean distance) to obtain probability distributions of training and test datasets belonging to each Meta-Domain. We treated these probability distributions as new domain vectors. Based on these domain vectors, we implemented the Distribution Alignment Assumption.
> > > >
> > > > The results are as follows:
> > > >
> > > > | Method  | Pearson | Spearman |
> > > > |----------|------------|------------|
> > > > | Meta-Domain Classifier           | 0.58330    | 0.66570  |
> > > > | KNN        | 0.40138    | 0.35429  |
> > > >
> > > > **The Meta-Domain Classifier significantly outperforms KNN.**
> > > >
> > > > However, the KNN method is also effective, which validates the rationality of our Meta-Domain vocabulary construction.
> > > >
> > > > At the same time, we would like to clarify that:
> > > > Embedding-based methods, in addition to having more limited context length compared to our method, still have several disadvantages.
> > > >
> > > > For different types of data, we used different clustering methods to construct Meta-Domains (line 126 - line 146).
> > > >
> > > > 1) For code, we directly identified its programming language without using an embedding model.
> > > > 2) For Chinese and English data, we used embedding models that output embeddings of different dimensions. Moreover, the semantic meaning of the same dimensions of different models' embeddings is  obviously different.
> > > >
> > > > Although these issues limit the generalizability of embedding-based methods, our proposed Meta-Domain Classifier is not constrained by the specific implementation methods used in constructing the meta-domain vocabulary. This also facilitates the introduction of new methods (except clustering) to construct more fine-grained Meta Domains.
> > > >
> > > > Thank you very much for taking the time to review our paper. If you have any further questions, we would be very happy to continue the discussion with you.
> > > >
> > > > Best regards,
> > > >
> > > > Authors of Paper 9648

---

### Official Review · Reviewer_9xLo · 2024-10-27

**Soundness:** 3
**Presentation:** 3
**Contribution:** 3
**Rating:** 8
**Confidence:** 3

**Summary:**

This paper introduces DOMAIN2VEC, a novel method for optimizing the data mixture ratio in LM pretraining by leveraging the concept of "Meta-Domains". DOMAIN2VEC decomposes datasets into a linear combination of these Meta-Domains, capturing essential features and maintaining a vocabulary for effective representation. By using a Meta-Domain Classifier, it generates domain vectors that facilitate the identification of optimal data mixtures without requiring retraining, based on the Distribution Alignment Assumption.

**Strengths:**

1. The authors decompose the dataset into domain vectors, representing attributes of datasets to balance its contributions, which introduces a training-free approach by for language models pretraining to find the optimal data mixture.

2. Experimental results indicate that DOMAIN2VEC can achieve better text generation capabilities and downstream task performance with lower computational costs.

3. This paper is well-written and provides detailed explanations from the theorical and practical aspacts.

**Weaknesses:**

1. The English requires further improvement and contains several minor errors. For example, line 80 is missing a period, and line 66 has an extra comma.

2. The authors claimed that "to the best of our knowledge, we are the first to propose DOMAIN2VEC, a novel concept for capturing the underlying features of datasets." There are works with similar ideas, particularly in the field of meta-learning, although they do not capsulate their approaches into the concept in the same way as this paper (e.g., `[1]`, `[2]`). The authors should discuss the relationship and differences between their work and these studies. Thus, I think it would be best to remove the statement of "the first".

Refs:

`[1]` Cross-Table Pretraining Towards a Universal Function Space for Heterogeneous Tabular Data.

`[2]` Dataset2Vec: Learning Dataset Meta-Features.

3. I have some doubts about the assumption. The authors state: "we introduce the Distribution Alignment Assumption (DA2), stating that lower validation loss can be achieved when the domain vector of training datasets aligns with the domain vector of the validation datasets." What if the domain vector of the training datasets is a subset of the domain vector of the validation datasets, or vice versa?

**Questions:**

See weakness.

---

> ### Author Response · Authors · 2024-11-22
> **Response to Reviewer 9xLo**
>
> Dear Reviewer 9xLo,
>
> Thanks for your very valuable review and recognition of our work! We will address your questions point by point.
>
> ## Q1: The authors should discuss the relationship and differences between their work and these studies (e.g., [1], [2]).
>
> Thank you very much for your specific suggestions! Dataset2Vec, XTFORMER, and our proposed Domain2Vec indeed share some similar ideas, establishing a latent space to model features in a domain. Thus, we supplemented the citations and discussions of prior studies (line 520 - line 523) to our paper and modified the corresponding statements. However, We believe that Domain2Vec is significantly different from the mentioned related works based on the following aspects.
>
> 1. The research questions and motivations are different and our paper mainly focuses on data mixture of LLMs pre-training.
> 2. The specific implementation of Domain2Vec is completely different from prior works (as we state in detail in section 2).
> 3. By obtaining representations of pre-training datasets in the same latent space, we can not only align datasets in the latent space (Domain2Vec +$DA^2$) but also model the relationship between latent space representations and model performance (Domain2Vec + RegMix).
>
> [1] Cross-Table Pretraining Towards a Universal Function Space for Heterogeneous Tabular Data.
>
> [2] Dataset2Vec: Learning Dataset Meta-Features.
>
> ## Q2: What if the domain vector of the training datasets is a subset of the domain vector of the validation datasets, or vice versa?
>
> We do not well understand this question and we will discuss this issue from two aspects.
>
> 1. A domain vector is a vector, rather than a set, thus it should be not possibly a "subset" of another domain vector.
> 2. Assuming your question is, what would the domain vector of the training dataset be if training dataset is a subset of the validation dataset? Our answer is: their domain vectors would be very similar (i.e., $\boldsymbol{\mathcal{v}}_1 \cdot \boldsymbol{\mathcal{v}}_2 \approx 1$) if domain 1 is randomly sampled (thus a subset) from domain 2 because the goal of Domain2Vec is just for calculating the features of a domain. When the sampling number becomes larger, $\boldsymbol{\mathcal{v}}_1$ becomes of less variance and finally stable as $\boldsymbol{\mathcal{v}}_2 $.
>
> Thanks for your time to read our response! We have updated our paper and fixed some typos. And we look forward to a further discussion with you.
>
> Best regard,
>
> Authors of Paper 9648

---

> ### Author Response · Authors · 2024-11-24
> **Looking forward to your reply!**
>
> Dear Reviewer 9xLo,
>
> This is a kindly reminder that the discussion period for our submission will end on Nov 26, 2024 (AOE). We have carefully addressed all concerns and questions raised in your review.
>
> Please let us know if there’s anything more we can clarify or elaborate on.
>
> Thank you so much for your insightful review and recognition of our work!
>
> Best regards,
>
> Authors of Paper 9648

---

### Official Review · Reviewer_me3R · 2024-11-02

**Soundness:** 3
**Presentation:** 2
**Contribution:** 2
**Rating:** 8
**Confidence:** 5

**Summary:**

This paper proposes a method for finding the mixing ratio of domains within a dataset. The method defines a concept of meta-domain that is represented as a vector that is a meta feature describing a domain. Given a dataset with some predefined domains, the method builds domain representations as a linear mixture of meta-domain vectors for each domain, then finds mixing ratios of domains as a function of domain vectors. The goal is to be more computationally efficient and scalable compared with prior works such as DoReMi. The paper relies on the Distribution Alignment Assumption (DA^2) that validation loss is minimized if the domain vector of training and validation sets align.

Section 2 introduces the Domain2Vec method. The method can be decomposed into the following steps:
- Construct vocabulary of meta-domains. A meta-domain, $D^*_j$, is a vector in the text embedding space. A vocabulary of 260 meta-domains is constructed from the centroids of KMeans in the embedding space of 5.2TBs of English, Chinese, and Code data.
- Create meta-domain classifiers. The method trains a text classifier that, given an input text, predicts the probability of it belonging to each meta-dataset. The classifier is a Qwen2-1.5B-base with a learnable head that is trained on 3k text samples from each meta-domain cluster.
- Define a domain vector and dataset vector. A domain vector, $v_i$, is an average of the probability vectors for text samples from a domain. A dataset is represented with a matrix of domain vectors, $V_{train}$.
- Compute the optimal mixing ratio $\bf{r}$ based on Eq. 5. That is, assuming the optimal mixing ratio is the one that maximizes the similarity between train and validation sets, and given the dataset matrix of the training set and domain vector of the validation set, the method finds the mixing ratio that maximizes the similarity between a mixed training set and the validation set.

---After rebuttal
Thank you for improving Eq 1. I recommend also fixing the minor issue of whitespace after thousand separator, e.g., in 3,000 and 1,000 on page 4. Maybe use {\small,}.

**Strengths:**

- The proposed method is ~3x more efficient than prior work, DoReMi.
- The method can efficiently adapt and scale as the number of domains within a dataset increases over time. In contrast, prior works would need to recompute the mixing ratios from scratch.

**Weaknesses:**

**Clarity**
- Line 18-19: Please consider defining or providing an intuition for Distribution Alignment Assumption in the abstract. Otherwise, the abstract is unclear.
- The notation is not clearly defined, for example, the dimensionalities, what is a vector and what is a scalar, can only be guessed from the equations rather than being clearly defined.
- Various concepts need to be clearly defined. For example, The meta-domain is not clearly defined and lines 118-119 are vague. Please consider clearly defining what is a vocabulary, features, and semantic features. Is a feature a vector? Is it an abstract concept? Is vocabulary a set of features? What is a semantic feature?
- On lines 56-57, the paper notes a downside of prior works as relying on a well-trained reference model which would imply the proposed method does not rely on such a model. However, the meta-domains rely on such a model as described in lines 134-147, specifically, `bge-small-en-v1.5` and `bge-small-zt-v1.5`. The meta-domain classifier also relies on a pretrained model, specifically, Qwen2-1.5B-base. Please consider clarifying the claim.

**Results**
- Table 3: The improvement in validation loss does not translate to any improvement in downstream performance. Specifically, the proposed method does not seem to be better than “Pile-CC only” that is a baseline with zero additional cost.
- Figure 5: Please consider adding more baselines such as DoReMi, a simple 50-50 mix of Pile and C4, and a baseline that deduplicates this simple mix.

**Minor**
- Line 80: missing period after “distinct features”.
- Please consider making a clear distinction between domains, $D_i$, and a training dataset, $D_{\text{train}}$. Right now the paper refers to both as datasets but defines a domain vector as a representation of $D_i$. So it makes more sense to also refer to it as a domain. Also the matrix $V_{\text{train}}$ can then be referred to as the dataset representation matrix or if flattened, a dataset vector. That way, the hierarchy of concepts, meta-domain, domain, and dataset, is more clearly separated.
- Line 162: belong -> belongs.
- There is a redundant space in all numbers with a thousand separate, right after the comma. Example: line 163, 3, 000. Please consider fixing this.
- Figure 2: the font size is too small, please consider reducing the columns/rows to the most important ones.
- Line 220: paer -> paper.
- Section 3.3 abuses the notation of $\mathcal{L}_\theta$ as it was previously defined as a function of $\bf{r}$ but now is a function of first $p, D_i^*$ and then just $\bf{p}$. Also, domain vector was previously denoted by $\bf{v}$ whereas now it is denoted by $\bf{p}$.
- Line 235: addictive -> additive.
- Line 267: what does “relative loss sizes” mean?

**Questions:**

- Figure 2: Please clarify how one would infer the claim on lines 172-173 from this figure that is “our classifier could reasonably distinguish some base features from different datasets.”.
- My understanding of Section 3.1 is that the proposed method relies on a predefined split of the dataset into domains. Is that correct? At the same time, the abstract says: “Domain2Vec … uses a Meta-Domain Classifier to decompose any given dataset into a domain vector that corresponds to a distribution over this vocabulary.”. This sentence could imply the method automatically identifies the domains. Please clarify the assumptions and requirements in the paper and particularly this sentence in the abstract.
- The notation of the loss in section 3.1 does not seem correct. Shouldn’t there be an additional inner argmin over $\theta$ given the mixing ratio $\bf{r}$? In other words, the model parameterized by $\theta$ will be different as one varies $\bf{r}$ but the Eq 4 would imply that it is independent.

---

> ### Author Response · Authors · 2024-11-22
> **Response to Reviewer me3R  [1/2]**
>
> Dear Reviewer me3R,
>
> Thanks for your insightful review! Q1 – Q3 are our responses to your questions and W1 – W5 are our responses to the weaknesses you have mentioned.
>
> ## Q1: How one would infer the claim on lines 172-173 from this figure that is “our classifier could reasonably distinguish some base features from different datasets.”?
>
> We clarify this question from the two following aspects:
>
> Firstly, in Lines 168-169, we reported that the Meta-Domain Classifier achieved a classification accuracy of 74.73% on the test set. It suggests that our classifier can distinguish the differences among the datasets and get a accurate result.
>
> Secondly, in Figure 2, we presented the decomposition results of different datasets by the Meta-Domain Classifier. For example, FreeLaw contains a large amount of data from Meta-Domain 232, while DM Mathematics contains a large amount of data from Meta-Domain 142. This indicates that FreeLaw and DM Mathematics have very different compositions, which aligns with our prior knowledge.
>
>
> ## Q2: My understanding of Section 3.1 is that the proposed method relies on a predefined split of the dataset into domains. Is that correct?
>
> There might be some misunderstandings. Section 3.1 describes the basic **task formulation**, which is to find the optimal mixture ratio from a given list of datasets (e.g. Github + ArXiv + Wikipedia). The task formulation is independent on any methods and keeps consistent across many other previous works like [1], [2], [3].
>
> However, Our method itself **never** relies on a predefined split of the dataset, we explain this question from training and inference stages:
>
> 1. In training stage, we train a Domain2Vec classifier based on the clustering result of an **unsupervised** method, kmeans.
> 2. In inference stage, we can adopt our Domain2Vec classifier to automatically identify an unseen domain, thus generate a vector of the domain.
>
> [1] [Data Mixing Laws: Optimizing Data Mixtures by Predicting Language  Modeling Performance](https://arxiv.org/pdf/2403.16952)
>
> [2] [RegMix: Data Mixture as Regression for Language Model Pre-training](https://arxiv.org/pdf/2407.01492)
>
> [3] [DoReMi: Optimizing Data Mixtures Speeds Up Language Model Pretraining](https://arxiv.org/pdf/2305.10429)
>
>
>
> ## Q3: Please clarify the $DA^2$ assumptions and requirements in the paper and particularly this sentence in the abstract.
>
> Thank you for pointing this issue out! We have updated the abstract to better clarify our assumptions. Specifically, the revised sentence becomes "... under the Distribution Alignment Assumption (DA2), which suggests that when the data distribution of the training set and the validation set is more aligned,  a lower validation loss is achieved." (line 16 - line 20)
>
>
>
> ## Q4: The notation of the loss in section 3.1 does not seem correct. Shouldn’t there be an additional inner argmin over given the mixing ratio.
>
> Thanks for your careful review！We have revised the notation of the loss in section 3.1 and other related notations in the updated version of our paper.

---

> ### Author Response · Authors · 2024-11-22
> **Response to Reviewer me3R  [2/2]**
>
> ## W1: The notation is not clearly defined, for example, the dimensionalities, what is a vector and what is a scalar, can only be guessed from the equations rather than being clearly defined.
>
> A1: In this paper, we strictly follow the recommended notation provided in the ICLR 2025 template.
>
> (see page 4 in https://www.overleaf.com/latex/templates/template-for-iclr-2025-conference-submission/gqzkdyycxtvt.pdf)
>
> For example, in our paper, $p$ is a scalar, $\boldsymbol{p}$ is a vector, and $\boldsymbol{V}_{train}$ is a Matrix.
>
> If you have any more questions about the notation somewhere, could you please point out which specific notation?
>
> ## W2: Please consider clearly defining what is a vocabulary, features, and semantic features. Is a feature a vector? Is it an abstract concept? Is vocabulary a set of features? What is a semantic feature?
>
> We are drawing an analogy between Domain2Vec and Word2Vec to help people in relevant fields better understand.
>
> **Vocabulary**: a set composed of Meta-Domains.
>
> **(Semantic) Features**(of a domain): a probability distribution on the vocabulary, which is computed by our Domain2Vec classifier. We use the term "semantic" here because our classifier is based on semantic embeddings.
>
> Thanks for your suggestions and we will add the definitions in our paper to help readers better understand our ideas.
>
> ## W3: Issues on the dependence on well-trained reference model.
>
> We think we have a strong response. There is a big difference between our embedding model and their reference models.
>
> Prior works trains a reference model (which is typically a smaller-size of the final language model) to help them find more valuable domains to learn during training. **When the training dataset changes, these reference models used in previous works need to be retrained.**
>
> Conversely, Domain2Vec adopts BGE and Qwen-2 models to compute domain vectors of different datasets, and they do not participate in the model training process.  The parameters of the models are frozen. And the models could be viewed as a tool to compute embeddings and they can be easily replaced by any other embedding models (such as a simple FastText).
>
> Overall, the procedure in our work does not reply such a reference model, and provides a much simpler and more efficient way to compute the optimal mixture ratio.
>
> ## W4: The improvement in validation loss seems not to translate to any improvement in downstream performance.  No better than “Pile-CC only” that is a baseline with zero additional cost.
>
> We clarify this issue from the following aspects:
>
> 1. Our work does not intend to research on the relationship between validation loss and downstream performance. Instead, we followed RegMix's experimental design and baselines (improvement in validation loss). We would like to emphasize that our contributions mainly lie in efficiency and scalability: in Table 3, Domain2Vec has achieved comparable downstream performance with only 0.26% of DoReMi's computational cost.
> 2. In RegMix's work,  it shows that validation loss on Pile-CC has the strongest correlation with the benchmarks, thus we follow and consider Pile-CC as validation set in this work. Interestingly, the better downstream performance achieved by "Pile-CC Only" actually confirms the correctness of our proposed Domain Alignment Assumption ($DA^2$).
>
> 3. Yes,  "Pile-CC only” is good when we regard Pile-CC as validation set. However, in a more practical scenario where validation set is somewhat else, we cannot manually find such a golden training set which can 100% correspond to validation set. However, we can use our proposed Domain2Vec to get a comparable performance with lowest cost.
>
> ## W5: For Figure 5, please consider adding more baselines such as DoReMi, a simple 50-50 mix of Pile and C4, and a baseline that deduplicates this simple mix.
>
> 1. Figure 5 aims to demonstrate that the data mixture found by Domain2Vec can reduce the loss on Pile-CC more quickly compared to the original data mixture in The Pile. In Table 3, we have indeed compared downstream performance with baseline, DoReMi. The results show that we achieved comparable downstream performance using only 0.26% of DoReMi's computational cost.
>
> 2. The experiments corresponding to Figure 5 only involve adjusting data mixture within The Pile's sub-datasets, so adding C4 doesn't seem to be a reasonable baseline.
>
> 3. Finally, data deduplication is fundamentally orthogonal to our proposed method. Typically, data deduplication and optimal mixture ratio discovery are two adjacent procedures in LLM pre-training. We believe that data deduplication can absolutely improve the performance and it does NOT conflict with our proposed method.
>
> Thank you very much for your careful review! We have updated the paper and corrected the typos you mentioned. We look forward to receiving your response!
>
> Best regards,
>
> Authors of Paper 9648

---

> ### Author Response · Authors · 2024-11-24
> **Looking forward to your reply!**
>
> Dear Reviewer  me3R,
>
> We are writing to kindly remind you that the discussion period for our submission is approaching its conclusion on Nov 26, 2024 (AOE). We have thoroughly responded to the comments and concerns you’ve raised.
>
> And if there are any points you’d like to discuss further or additional input you’d like from us, we would be delighted to assist before the deadline.
>
> We truly appreciate your time and effort in reviewing our work!
>
> Best regards,
>
> Authors of Paper 9648

---

> > ### Comment · Reviewer_me3R · 2024-11-26
> >
> > I thank authors for their response. Some of my concerns are addressed. I raise my rating. One point regarding the notation, specifically the dimensionality of Equation 1 is confusing because D has not been defined as a scalar or vector properly.

---

> > > ### Author Response · Authors · 2024-11-27
> > > **Response to Reviewer me3R**
> > >
> > > Dear Reviewer me3R,
> > >
> > > Thanks for your reply of our response.
> > >
> > > Regarding the notation in Equation 1, we have updated our manuscript (line 118 - line 126) to improve the clarification of Equation 1. Thank you again for acknowledging our work. We would like to continually improve our work if you have any further comment.
> > >
> > > Best regards,
> > >
> > > Authors of Paper 9648

---

### Official Review · Reviewer_s3iG · 2024-11-04

**Soundness:** 3
**Presentation:** 3
**Contribution:** 2
**Rating:** 3
**Confidence:** 4

**Summary:**

The paper proposes a way to featurize datasets by computing their similarity to a set of "meta-domains", which are essentially the result of a k-means clustering of text embeddings. The authors then show how to use their featurizations / domain vectors to optimize dataset compositions, i.e., what weights to assign to the components of a combined dataset (such as arXiv, Wikipedia, etc.).

**Strengths:**

- The paper studies an important problem
- The proposed method is interesting and computationally efficient
- The paper is clearly written

**Weaknesses:**

My main reservation is that the proposed method offers little to no gains empirically. In particular, the authors conducted experiments where they trained language models on multiple different training sets and then measured the performance of the resulting models on a range of downstream tasks. I think this is a good experimental setup. Unfortunately, the results (Table 3) show that the proposed method offers little to no gains. In particular, the best Domain2Vec variant is worse than RegMix (0.483 vs 0.489) and worse than only training on Pile-CC alone (0.483 vs 0.487). The proposed method Domain2Vec is computationally cheaper than RegMix (9.7e16 flops vs 3.5e18 flops). But it is not clear that this difference is relevant because both flops counts are smaller than the cost of training the language model (6 N D gives 1.2e20 flops). Negative results do not necessarily mean that a submission must be rejected. But the paper should frame the results appropriately as negative and discuss the comparison to the aforementioned baselines more clearly.

In addition, the paper does not compare to state-of-the-art pre-training datasets such as RefinedWeb (https://arxiv.org/abs/2306.01116), FineWeb (https://arxiv.org/abs/2406.17557), and DCLM (https://arxiv.org/abs/2406.11794).

**Questions:**

- Line 65: small type (",,")
- Line 99: small typo ("to for vectorizing")

---

> ### Author Response · Authors · 2024-11-22
> **Response to Reviewer s3iG**
>
> Dear Reviewer s3iG,
>
> Thanks for your careful review! We will reply to your questions one by one and hope to solve your concern.
>
> ## Q1: The proposed method seems to offer little to no gains empirically.
>
> 1. Domain2Vec achieved comparable downstream performance with only 0.26% of DoReMi's computational cost. Extremely high efficiency while maintaining good performance is Domain2Vec's advantage. This has been acknowledged by Reviewers me3R, 9xlo, and tRxw.
>
> >  Reviewer `me3R`: The proposed method is ~3x more efficient than prior work, DoReMi. The method can efficiently adapt and scale as the number of domains within a dataset increases over time.
>
> >  Reviewer `9xlo`: Experimental results indicate that DOMAIN2VEC can achieve better text generation capabilities and downstream task performance with lower computational costs.
>
> >  Reviewer `tRxw`: The results demonstrate that the proposed method is effective.
>
> 2. Our method's efficiency becomes particularly significant when the training dataset changes and scales up. For example, when we add a new training dataset, RegMix needs to give up everything, and re-collect approximately $|\mathcal{D}|^2$ ($|\mathcal{D}|$ denotes the number of datasets) data mixture to fit the linear regression model. In contrast, Domain2Vec only needs to use the Meta-Domain classifier to convert the new dataset into a vector representation. Note that in practical scenarios, the number of datasets could be significantly more than 20 (e.g. 200). It's clear that RegMix's computational cost increases quadratically with the number of datasets, while Domain2Vec increases linearly.
>
> 3. Our method provides a novel perspective on optimizing data mixture, and Domain2Vec can **naturally integrate with prior work** (e.g., Domain2Vec + RegMix), this further suggest the scalability of our proposed method.
>
>
> ## Q2: The paper does not compare to state-of-the-art pre-training datasets such as RefinedWeb, FineWeb, and DCLM.
>
> The main research scope of our work is to find the optimal data mixture among multiple pretraining datasets, rather than constructing a better pretraining dataset. **Our proposed method is orthogonal to dataset sources**. For example, Domain2Vec can also adjust the datat mixture of different domains within RefinedWeb. In this paper, to align with prior works, such as DoReMi, RegMix, we primarily selected The Pile, C4, and Knowledge-Pile as our pre-training datasets.
>
> We appreciate your thorough review and insightful questions. We hope we address each point you mentioned in detail. And we look forward to your further feedback.
>
> Best regard,
>
> Authors of Paper 9648

---

> > ### Comment · Reviewer_s3iG · 2024-11-25
> > **Computational efficiency etc.**
> >
> > Thank you for describing your perspective in more detail. I still have the following questions:
> >
> > - I agree that Domain2Vec is more efficient than DoReMi, but is this the most relevant comparison? The results in the submission show that RegMix is more efficient than DoReMi. When comparing to prior work, the comparison to the best prior work is particularly relevant.
> >
> > - As I mentioned in my original review, it is not clear to me that the computational advantage over RegMix is relevant because the computational cost, at least based on flops count, is more than 10x smaller than training the model. Is this correct? If so, the training mix optimization with RegMix is not the bottleneck here.
> >
> > - If I understand the results correctly, at the 1B scale (Table 3), Domain2Vec offers no advantage in aggregate model performance over training on a single data source only (Pile-CC). Is this correct? Training on Pile-CC alone also requires no cost for optimizing the training mix. It would be helpful to see results in a regime where Domain2Vec results in a better model.
> >
> > Finally, I understand the author's point that their method is orthogonal to work on dataset sources. Nevertheless, I find these comparisons important because performing experiments on potentially outdated datasets may miss important phenomena that only occur near the state of the art.

---

> ### Author Response · Authors · 2024-11-24
> **Looking forward to your reply!**
>
> Dear Reviewer s3iG,
>
> Thank you again for your time and effort in reviewing our work! As a reminder, the discussion period for our submission will conclude on Nov 26, 2024 (AOE). We have carefully addressed the questions and concerns raised in your review and sincerely look forward to your valuable feedback.
>
> If you have any additional questions or require further clarification, we would be more than happy to engage and provide prompt responses during this period.
>
> Best regards,
>
> Authors of Paper 9648

---

> ### Author Response · Authors · 2024-11-25
> **Response to Reviewer s3iG  [1/2]**
>
> Dear Reviewer s3iG,
>
> Thanks for your feedback of our response.
>
> ## Q1: I agree that Domain2Vec is more efficient than DoReMi, but is this the most relevant comparison? The results in the submission show that RegMix is more efficient than DoReMi. When comparing to prior work, the comparison to the best prior work is particularly relevant.
>
> In Table 3, we also compared with RegMix. **Correspondingly, Domain2Vec's computational cost is only 2.76% of RegMix's.** Additionally, as we mentioned in our initial response to your review, "RegMix's computational cost increases quadratically with the number of datasets, while Domain2Vec increases linearly. This demonstrates that our method is more efficient than RegMix. It is also notable that, while RegMix requires training language models, Domain2Vec only needs to perform inference. This gives Domain2Vec significant advantages in terms of code implementation complexity and GPU memory consumption. Please refer to the following Q2 for more explanations and comparisons with our method and RegMix.
>
>
> ## Q2: As I mentioned in my original review, it is not clear to me that the computational advantage over RegMix is relevant because the computational cost, at least based on flops count, is more than 10x smaller than training the model. Is this correct? If so, the training mix optimization with RegMix is not the bottleneck here.
>
> No, it is not correct. Taking training a 1B parameter model for 25B tokens as an example (following RegMix), the Flops required to train this model is approximately 1.5e+20 (N = 1B, D=25B). The following table shows how the FLOPs of Domain2Vec and RegMix change with the number of training datasets. When the number of training datasets reaches 200, RegMix's FLOPs already exceed those required for training the model itself.
>
>
> | Number Of Training Datasets | Domain2Vec | RegMix     |
> | --------------------------- | ---------- | ---------- |
> | 17 (The Pile)               | 9.6600e+16 | 3.5000e+18 |
> | 100                         | 5.4203e+17 | 6.8359e+19 |
> | 200                         | 1.0787e+18 | 2.7344e+20 |
> | 500                         | 2.6887e+18 | 2.7344e+20 |
> | 1000                        | 5.3720e+18 | 1.7090e+21 |
> | 2000                        | 1.0739e+19 | 2.7344e+22 |
>
> Note that over 1K training datasets in LLM training is quite possible. For example, your mentioned Fineweb[1] has over **10K** dumps with inconsistent distributions. Another example is Qwen-2.5 [2] and Llama-3 [3], which are trained on over 15 trillion tokens—approximately 100 times the size of The Pile dataset.
>
> Our method provides a universal representation of pre-training datasets, which focuses on the **scalability** and efficiency of pre-training data mixture experiments. The scalability of Domain2Vec is reflected in: Domain2Vec establishes a latent space representation of datasets. Therefore, any pre-training data can be mapped into this space. **Experiments conducted in the latent space remain consistent regardless of changes in pre-training datasets.** In contrast, RegMix performs experiments at the dataset level, requiring all previous experimental results to be discarded and new experiments to be conducted when datasets change (such as, adding a new datasets, improving the quality of some datasets).
>
> Last but not least, **our method does not replace RegMix but can enhance its scalability and efficiency using our Domain2Vec.** Please refer to the "Domain2Vec + RegMix" section in our paper.
>
> [1] [The FineWeb Datasets: Decanting the Web for the Finest Text Data at  Scale](https://arxiv.org/pdf/2406.17557)
>
> [2] [Qwen2.5: A Party of Foundation Models]( https://qwenlm.github.io/blog/qwen2.5/)
>
> [3] [The Llama 3 Herd of Models](https://arxiv.org/pdf/2407.21783)

---

> ### Author Response · Authors · 2024-11-25
> **Response to Reviewer s3iG [2/2]**
>
> ## Q3: If I understand the results correctly, at the 1B scale (Table 3), Domain2Vec offers no advantage in aggregate model performance over training on a single data source only (Pile-CC). Is this correct? Training on Pile-CC alone also requires no cost for optimizing the training mix. It would be helpful to see results in a regime where Domain2Vec results in a better model.
>
> We would like to clarify this issue from the following aspects:
>
> 1. Our work does not intend to research on the relationship between validation loss and downstream performance. Therefore, we followed RegMix's experimental design and baselines.
> 2. RegMix finds that validation loss on Pile-CC has the strongest correlation with the benchmarks they evaluates. Thus, we follow and consider Pile-CC as validation set in our experiments. Interestingly, the better downstream performance achieved by "Pile-CC Only" actually confirms the correctness of our proposed Domain Alignment Assumption ($DA^2$).
> 3. Most importantly, "Pile-CC only” is good when we regard Pile-CC as validation set. **However, in a more practical scenario where validation set is somewhat else, we cannot manually find such a golden training set which can 100% correspond to validation set.**. However, we can use our proposed Domain2Vec to get a comparable downstream performance with lowest cost by mixing datasets from different sources.
>
> Lastly, thank you for understanding that our method and datasets are orthogonal. Due to computational resource constraints, and to ensure fair comparison with prior works, we chose to use the Pile. However, in Table 1, we also conduct experiments on the Knowledge Pile (A recent high quality pretraining dataset). We consider experiments on higher-quality datasets as our future work.
>
> We hope that our response can address your concern, and we look forward to your further feedback.
>
> Best regards,
>
> Authors of Paper 9648

---

> ### Author Response · Authors · 2024-11-29
> **Kind reminder for your feedback**
>
> Dear Reviewer s3iG,
>
> We are very grateful for the opportunity to improve our work through your valuable feedback. As the discussion period is near the end, this is a kind reminder for your further feedback.
>
> If you think our response has addressed your concerns and questions, we sincerely hope you might consider raising your score. Should you have any further inquiries or require additional clarification, we warmly welcome your questions and are eager to provide comprehensive responses.
>
> Best regards,
>
> Authors of Paper 9648

---

> ### Author Response · Authors · 2024-12-02
> **Looking forward to your reply!**
>
> Dear Reviewer s3iG,
>
> We are very grateful for the opportunity to improve our work through your valuable feedback. As the discussion period is near the end, this is a kind reminder for your further feedback.
>
> If you think our response has addressed your concerns and questions, we sincerely hope you might consider raising your score. Should you have any further inquiries or require additional clarification, we warmly welcome your questions and are eager to provide comprehensive responses.
>
> Best regards,
>
> Authors of Paper 9648

---

> ### Author Response · Authors · 2024-12-03
> **Looking forward to your feedback!**
>
> Dear Reviewer s3iG,
>
> We greatly appreciate the opportunity to enhance our work based on your invaluable feedback. **As the discussion period draws to a close,  this is a kind reminder for your further feedback.**
>
> If our response has addressed your concerns and questions, we sincerely hope you might consider raising your score. Should you have any further inquiries or require additional clarification, we warmly welcome your questions and are eager to provide further responses.
>
> Warm regards,
>
> Authors of Paper 9648

---

### Author Response · Authors · 2024-11-22
**Global Response to All Reviewers**

We sincerely thank the reviewers for their valuable feedback and insightful comments. We are glad that the reviewers have acknowledged many aspects of our work:

1. The research question of this paper is important (reviewer `s3iG`).
2. Our proposed method is intuitive, computationally efficient, and effective in obtaining optimal data mixture (reviewer `s3iG`, `me3R`, `9xLo`,`tRxw`).
3. This paper is clearly written and provides detailed explanations from the theoretical and practical aspects (reviewer `s3iG`, `9xLo`).

Also, We have carefully considered each point of all the reviews and have made the necessary revisions to enhance the clarity and quality of our manuscript. The summary of our main revision is as below:

1. We supplemented some descriptions about related works in Line 519-522, based on the suggestions of reviewer `9xLo`.
2. We added some explanations to better reflect the definition of some terms (e.g., $DA^2$, vocabulary, features), based on the suggestions of reviewer `me3R`.
3. We modify some notations in section 3.1-3.3 and re-plot Figure 2 to make our paper clearer and easy to follow, based on the suggestions of reviewer `me3R`
4. We fixed all the mentioned typos (notation and text) and grammar errors in the paper, based on the suggestions of reviewer `s3iG`, `me3R`, `9xLo`.

All the revised sentences are highlighted in red. Besides, we provide detailed responses to each of the reviewers' comments as below.

We appreciate all the reviewers' constructive feedback, which has significantly improved our work. We hope the revisions meet your expectations and look forward to your further feedbacks.

---

### Meta-Review · Area_Chair_zTtN · 2024-12-20

**Metareview:**

This paper introduces DOMAIN2VEC, a novel approach for optimizing the data mixture ratio in language model (LM) pretraining by leveraging the concept of Meta-Domain. DOMAIN2VEC decomposes datasets into a linear combination of Meta-Domains, capturing essential features while maintaining a vocabulary for effective representation. Using a Meta-Domain Classifier, it generates domain vectors that enable the identification of optimal data mixtures without requiring retraining, based on the Distribution Alignment Assumption.

Overall, while the proposed method shows potential, the current submission faces some issues. 1) The paper lacks clarity in its explanations and suffers from poor English, which is pointed out by several reviewers. Addressing these issues will likely require substantial revisions to the text. 2) The experiments are conducted on older datasets rather than large-scale and state-of-the-art (SoTA) datasets. This is a critical limitation emphasized by one reviewer, as conclusions drawn from outdated datasets may miss important phenomena or trends that are only observable with recent SoTA data.

Considering the current acceptance rate and these limitations, the paper is not yet ready for publication. We recommend that the authors address these concerns comprehensively, as outlined in the reviewers' feedback, to improve the clarity and rigor of the work for future submissions.

**Additional Comments On Reviewer Discussion:**

Here I mainly list the unresolved concerns from Reviewer s3iG, and also list the common issue by other reviewers.

（1）Lacking experiments on large and recent SoTA datasets (Reviewer s3iG)
The authors claim they use the Pile dataset for fairness, and leave the experiments on large and recent SoTA datasets as their future work.

（2）Efficiency compared with Domain2Vec and RegMix  (Reviewer s3iG)
The authors provide some explanations, and should address the concerns.

（3）Poor clarity and English issues (most Reviewers)
The authors provide some explanations and promise careful polishing.

My points:

1) The paper lacks clarity in its explanations and suffers from poor English, which is pointed out by several reviewers. Addressing these issues will likely require substantial revisions to the text.

2) The experiments are conducted on older datasets rather than large-scale and state-of-the-art (SoTA) datasets. This is a critical limitation emphasized by one reviewer, as conclusions drawn from outdated datasets may miss important phenomena or trends that are only observable with recent SoTA data. I actually partially agree with this point.

---

### Decision · Program_Chairs · 2025-01-22

Reject